# Longitudinal contrast in Turbulence along a $\sim 19°$S section in the Pacific and its consequences on biogeochemical fluxes

Pascale Bouruet-Aubertot[1], Yannis Cuypers[1], Andrea Doglioli[2], Mathieu Caffin[2], Christophe Yohia[2], Alain de Verneil[2], Anne Petrenko[2], Dominique Lefèvre[2], Hervé Le Goff[1], Gilles Rougier[2], Marc Picheral[3], and Thierry Moutin[2]

[1]Sorbonne Université- UPMC Univ. Paris 06- LOCEAN, France
[2] Aix Marseille Univ, Université de Toulon, CNRS, IRD, MIO UM 110 , 13288, Marseille, France
[3] LOV, Villefranche sur mer, France

**Correspondence:** P. Bouruet-Aubertot (pascale.bouruet-aubertot@upmc.fr)

**Abstract.** Microstructure measurements were performed along the OUTPACE longitudinal transect in the tropical Pacific (Moutin and Bonnet, 2015). Small-scale dynamics and turbulence in the first 800m surface layer were characterized based on hydrographic and current measurements at fine **vertical** scale and turbulence measurements at cm scale using a vertical microstructure profiler. The possible impact of turbulence on biogeochemical budgets in the surface layer was also addressed in this region of increasing oligotrophy to the East. The dissipation rate of turbulent kinetic energy, $\epsilon$, showed an interesting contrast along the longitudinal transect with **stronger** turbulence in the West, i.e. the Melanesian Archipelago, compared to the East, within the South Pacific Subtropical Gyre, **with a variation of** $\epsilon$ **by a factor of 3 within** $[100m - 500m]$. The layer with enhanced turbulence decreased in vertical extent traveling eastward. This spatial pattern was correlated with the energy level of the internal wave field, higher in the West compared to the East. The difference in wave energy mostly resulted from enhanced wind power input into inertial motions in the West. Moreover, three long duration stations were sampled along the cruise transect, each over three inertial periods. The analysis from the western long duration station gave evidence of an energetic baroclinic near-inertial wave that was responsible for the enhanced $\epsilon$, observed **within a 50m-250m layer**, **with a value of** $810^{-9} W kg^{-1}$**, about 8 times larger than at the eastern long duration stations**. Averaged nitrate turbulent diffusive fluxes **in a 100-m layer below the top of the nitracline** were about twice larger **west of 170W** due to the higher vertical diffusion coefficient. In the **photic** layer, **the depth-averaged** nitrate turbulent diffusive flux **strongly decreased eastward with an averaged value of** $11 \mu mol m^{-2} d^{-1}$ **West of 170W to be compared with the** $3 \mu mol m^{-2} d^{-1}$ **averaged value East of 170W. Contrastingly phosphate turbulent diffusive fluxes were significantly larger in the photic layer**. This input may have an important role in sustaining the development of N$_2$-fixing organisms that were shown to be the main primary contributors to the biological pump in the area. The time-space intermittency of mixing events, intrinsic to turbulence, was underlined but its consequences on micro-organisms would deserve a dedicated study.

# 1 Introduction

The subtropical South Pacific is one of the main oceanic deserts characterized by an increasing oligotrophy to the East **and the center of the gyre**. A 43-day long cruise, the OUTPACE experiment, was performed in this region, along an $\sim 19°$S longitudinal transect, during the 2015 austral summer in order to characterize the biological pump and its coupling with dynamical processes (Moutin et al., 2017). In addition to the trophic gradient the OUTPACE transect is also characterized by a longitudinal contrast in dynamics between the 'energetic' Melanesian Archipelago (MA) and the 'quiet' South Pacific Subtropical Gyre (SPSG) (e.g., Rousselet et al., 2017). Hence the OUTPACE experiment provides a unique opportunity to focus on physical and biological interactions (e.g., Rousselet et al., 2017) that may prove crucial in understanding biological pump functionning (e.g., Guidi et al., 2012; Ascani et al., 2013). The influence of the mesoscale and submesoscale circulations on the spatial distribution and transport was detailed by Rousselet et al. (2017). In particular they showed the strong impact of fronts on the spatial distribution of bacteria and phytoplancton. A detailed study of an anomalous surface bloom event by de Verneil et al. (2017) revealed instead the main impact of mesoscale advection. At smaller-scales three dimensional turbulence may have a strong impact on the biological pump through the input of nutrients into the photic layer and more generally in enhancing, in the stratified ocean, vertical transports through turbulent diffusion (e.g., Ledwell et al., 2008).

The level of turbulence is almost unknown in the OUTPACE area. To our knowledge, the only microstructure measurements were performed in the western part of the subtropical South Pacific during the Malaspina expedition (Fernández-Castro et al., 2014, 2015) as part of an extensive microstructure survey in the tropical and subtropical oceans. For the leg done in the OUTPACE region, the averaged $\epsilon$ **below the mixed layer down to** $\sim 300m$ **depth** was $\sim 10^{-8}\, Wkg^{-1}$, well above the typical background dissipation rate for open ocean. Indirect estimates of $\epsilon$ based on ARGO floats data fall in the same range as Fernández-Castro et al. (2014) as shown by Whalen et al. (2012). This study based on the global scale ARGO floats dataset also revealed that the South subtropical Pacific is one of the most undersampled area. At the larger scale of the South Pacific ocean, the equatorial zone is well-known as a hotspot for turbulence where shear instability prevails as a result of the strongly sheared current system (e.g., Gregg et al., 1985; Sun et al., 1998; Richards et al., 2015; Smyth et al., 2013). At subtropical latitudes, where the background shear is lower, internal waves are expected to play a major role on the onset of turbulence in the stratified interior. Global maps of energy flux show enhanced semi-diurnal tide energy conversion in the western part of the subtropical South Pacific **(Alford and Zhao, 2007a, Fig.9b). The annual mean energy flux into inertial motions is enhanced at mid-latitudes in all ocean basins with also a SE oriented track in the Pacific from the Equator to 40S and within** $\sim 180°$**E -**$160°$**W longitude in the OUTPACE region** (Alford and Zhao, 2007a, Fig.9a). The latter process is subject to seasonal variations especially in subtropical regions where the generation of energetic baroclinic near-inertial waves is favored during the cyclone season (e.g., Liu et al., 2008).

The contribution of the biological pump in the OUTPACE region to the main C, N and P biogeochemical cycles was one of the main purposes of the OUTPACE project (Moutin et al., 2017). Moutin et al. (2018) built a first-order budget at daily scale of these main elements while Caffin et al. (2018) focused on the role of $N_2$ fixation. $N_2$ fixation was evidenced as the dominant

process involved in the N cycle in regions where *Trichodesmium* dominate. The input of nitrate through turbulent diffusion was found to make a negligible contribution in the photic layer as a result of a very deep nitracline. This rose the question of the available source of other nutrients in the photic layer that could sustain the development of $N_2$-fixing organisms, the main primary contributors to the biological pump in the area (Caffin et al., 2018).

The purpose of this paper is to characterize **the spatial variability of turbulence** along the OUTPACE transect with microstructure measurements performed at both one-day short duration stations and at long duration stations lasting three inertial periods. The idea is also to provide insights into the main mechanisms responsible for the observed turbulence with a focus on long duration stations that allow a characterization of the internal wave field. How this small-scale dynamics influences biogeochemical fluxes is another issue that is eventually addressed.

## 2    Data and methods

The OUTPACE cruise took place in early 2015 from February 18th to April 3rd onboard the French oceanographic research vessel l'Atalante (Moutin et al., 2017). A set of 15 short duration stations (SD) over 24h as well as 3 long duration stations (LD) over three inertial periods **(the inertial period being of $\sim 36h$)** were performed along an almost zonal transect starting from west of New Caledonia and ending near Tahiti (Fig. 1).

### 2.1    CTD and LADCP

Conductivity-Temperature-Depth (CTD) measurements were performed on a rosette using a SeaBird SBE 9plus instrument. Data were averaged over 1-m bins to filter out spurious salinity peaks using Sea-Bird electronics software. Simultaneously, currents were measured from a 300 kHz RDI Lowered broadband acoustic Doppler current profiler (LADCP). LADCP data were processed using the Visbeck inversion method (Visbeck, 2002) and provided vertical profiles of horizontal currents at

8 m resolution. **These measurements were performed at all stations with a typical 3 hours time interval between each deployment**. In addition the ship was equipped with two SADCPs, RDI Ocean Surveyors with frequencies 150kHz and 38kHz yielding processed currents averaged over 2 min time interval and with vertical bins of 8m and 24m respectively. **Shear was computed using finite differences with current vertical profiles interpolated over a $1-m$ vertical grid with an estimated noise level of $5.10^{-4}s^{-1}$.**

### 2.2    Microstructure measurements with VMP1000

Microstructure measurements were collected using a vertical microstructure profiler, 'VMP1000' (Rockland Scientific). This tethered profiler was equipped with microstructure sensors, two shear sensors and one temprature sensor, as well as with Sea-Bird temperature and conductivity sensors and a high frequency fluorometer. A total number of 123 profiles were performed with repeated profiles at LD stations **($\sim$ 30 profiles over 3 inertial periods)** and at least one profile at each SD station except

**at SD13 (see Table 1 for further details)**. The dissipation rate of turbulent kinetic energy ($\epsilon$) was inferred from centimeter-scale shear measurements. The vertical wavenumber shear spectrum was computed within the inertial range, typically within

meter to centimeter scales. The experimental spectrum was next compared to the empirical spectrum, the Nasmyth spectrum (Nasmyth, 1970), which allowed validation of the estimate of $\epsilon$ (e.g. for a detailed description, Ferron et al., 2014). **Shear measurements were processed using the routines developed by Rockland Scientific. Specific noise removal procedures were applied with the spikes in the shear data first removed and spectral coherence between the shear sensors and the accelerometers used to remove vibrational contamination. The first 20m below the surface were not considered to avoid any contamination from the ship wake as well as the 20m at the end of the profile because of the decreasing vertical velocity there. More generally $\epsilon$ values were excluded when the vertical velocity gradient of the VMP was larger than $2.5 10^{-2} s^{-1}$. The averaged $\epsilon$ from the two shear probes was taken provided that the ratio between the two estimates was smaller than 2, otherwise the $\epsilon$ value with the smallest depth variation (compared to the neighbouring upper and lower $\epsilon$ values) was considered.** $\epsilon$ was computed over a $1m$ depth interval, then a $8m$ moving average was applied on this signal. The estimated noise level is $5 \times 10^{-11} W kg^{-1}$ **following Ferron et al. (2014).**

## 2.3 Diffusivity estimates

The diapycnal diffusivity, $K_z$, is commonly inferred from the kinetic energy dissipation rate using the Osborn (1980) relationship:

$$K_z = \Gamma \epsilon N^{-2} \tag{1}$$

where $\Gamma$ is a mixing efficiency defined as the ratio between the buoyancy flux and the dissipation rate, $\Gamma = -\frac{g}{\rho_0} \frac{\overline{\rho' w'}}{\epsilon}$ with $w'$ and $\rho'$ the vertical velocity and density fluctuations, and $N$ the buoyancy frequency, inferred from the sorted density profile in order to avoid spurious negative values associated with overturns, $N = \sqrt{-\frac{g}{\rho_0} \frac{d\rho_{sorted}}{dz}}$, with a $8-m$ moving average then applied on this signal. $\Gamma$ was generally set to 0.2 until the recent findings of Shih et al. (2005) and Bouffard and Boegman (2013). These authors found a decrease of $\Gamma$ for increasing turbulence intensity, $I$, defined as:

$$I = \epsilon/(\nu N^2) \tag{2}$$

where $\nu$ is the molecular viscosity, $\nu = 1.2 \times 10^{-6} m^2 s^{-1}$. In term of timescales, $I$ is the ratio of the square of the Kolmogorov time scale, namely the dissipation time scale of eddies at the Kolmogorov scale ($\sqrt{\nu/\epsilon}$), and the buoyancy time scale ($1/N$). Shih et al. (2005) showed in a numerical study that the Osborn relationship overestimated $K_z$ when $I > 100$ and proposed a new parameterization of $K_z$ for this regime. A few years later Bouffard and Boegman (2013) proposed a refined parameterization of $K_z$ including in-situ microstructure measurements in lakes as well. They defined different regimes with the following formulations for $K_z$:

- $K_z = 10^{-7} m^2 s^{-1}$ within the diffusive sub-regime, $I < 1.7$

- $K_z = \frac{0.1}{7^{1/4}} \nu I^{3/2}$ within the buoyancy controlled sub-regime, $I$ within $[1.7; 8.5]$

- $K_z = 0.2 \nu I$, i.e. the Osborn relationship within the intermediate regime, $I$ within $[8.5, 400]$

– $K_z = 4\nu I^{1/2}$ within the energetic regime, $I > 400$

Note that for the OUTPACE datatset where most $I$ values are smaller than 100, the $K_z$ values inferred from the Bouffard and Boegman parameterization that is applied here do not differ signficantly from those inferred from the Osborn relationship.

## 2.4 Internal forcing: estimates of internal tide generating force and wind power input into inertial motions

The internal tide generation is inferred from the depth integrated generating force following the linear approximation (e.g., Baines, 1982) that reads:

$$\|\boldsymbol{F}\| = \int \frac{N^2}{\omega} \frac{z \|\boldsymbol{Q}.\nabla h\|}{h^2} dz$$

where $N$ is inferred from the World Ocean Atlas monthly climatology, $\omega$ is the tidal frequency, and $\|\boldsymbol{Q}\|$ is the barotropic tidal flux and $h$ the bottom depth. The barotropic tidal flux is inferred from the $1/30^o \times 1/30^o$ global inverse tidal model
TPXO (Egbert and Erofeeva, 2002) for two main constituents, the diurnal K1 and the semi-diurnal M2.

The generation of baroclinic near-inertial waves occurs through inertial pumping at the base of the mixed layer (e.g. Gill, 1984; Price, 1984). Insight on possible generation of baroclinic near-inertial waves is estimated from the wind-work on inertial oscillations following Alford (2003):

$$F_f = -\rho \frac{\tau_f^2}{rH}; \tag{3}$$

where $\tau_f^2$ is the square of the wind stress at the inertial frequency, $r$ is the damping of near-inertial motions in the mixed layer as a result of baroclinic near-inertial wave radiation expressed as a function of the inertial frequency: $r = 0.15f$ and $H$ the mixed layer depth. The wind stress was inferred from numerical simulations over the time period of the cruise (Skamarock et al., 2005) and the mixed layer depth from the seasonal climatology.

## 2.5 Biogeochemical turbulent diffusive fluxes

The vertical component of nitrate and phosphate turbulent diffusive fluxes were computed at all stations, using the diapycnal diffusivity, $K_z$, inferred from microstructure measurements:

$$F_{NO3,PO4} = -K_z \partial_z c_{NO3,PO4} \tag{4}$$

where $c_{NO3}$ and $c_{PO4}$ are the nitrate and phosphate concentrations. Measurements were performed daily at 12 depths in the first 200m using standard colorimetric procedures (see Caffin et al., 2018, for further details). The quantification
limits were $50\mu mol.m^{-3}$. At LD stations where 6 profiles were obtained, the quantification limit for the mean concentration dropped down to $20\mu mol.m^{-3}$ (i.e. $50/\sqrt{6}$). Concentrations values below the threshold for quantification were set to NaN. Vertical concentration profiles were interpolated over a $1m$ vertical grid and a 10-m moving average was next applied. The top of the nitracline was defined as the depth where nitrate concentration is zero based on an extrapolation from the last detectable concentration ($> 50\mu mol.m^{-3}$) assuming a constant vertical gradient above this depth

(see Moutin et al., 2018). It is only at long duration stations that the top of the nitracline was defined in isopycnal coordinates: this allows to get rid of a varying depth of the nitracline because of vertical displacements of isopycnals induced by internal waves (see Caffin et al., 2018). The minimum turbulent diffusive flux was estimated from the threshold concentration with the molecular $K_z$ value ($K_z \sim 10^{-7}m^2s^{-1}$): $0.4\mu mol m^{-2}d^{-1}$ **at SD stations and** $0.17\mu mol m^{-2}d^{-1}$ **at LD stations where the mean concentration profile was taken to compute the flux. A reference profile was also defined in order to compare the nitrate diffusive flux variations along the OUTPACE within a 100m layer below the top of the nitracline. To do so vertical profiles of concentration were first shifted vertically so that the reference depth matched with the top of the nitracline and the mean vertical** $c_{NO3}$ **profile within the nitracline was inferred from this set of re-scaled concentration profiles.** $K_z$ **profiles were as well rescaled onto this vertical grid and a mean** $K_z$ **profile inferred. The relative contributions of the variations in** $K_z$ **and that of** $\partial_z c_{NO3}$ **with respect to the total variations of the flux were also inferred.**

## 3 Spatial pattern of turbulence

### 3.1 Overview

An overview of the spatial pattern of turbulence is given with depth-averaged values of $\epsilon$ and $K_z$ below $100m$ **depth** at each station (Fig.1). Depth-averaged dissipation rates, $<\epsilon>$, vary within an order of magnitude within $[10^{-9.5}; 10^{-8.5}]Wkg^{-1}$. The highest values are observed West of **170W**, in the shallower part, while the lowest values are observed East of **170W**, in the deeper part. The same contrast is retrieved on $<K_z>$ with values ranging within $[10^{-5.8}; 10^{-4.8}]m^2s^{-1}$. The western part of our study area which shows the highest turbulence level is also the region where the most intense velocities are observed as illustrated with altimetry-derived currents produced by AVISO along the RV l'Atalante cruise path (Fig.2a). The vertical section of the total velocity modulus inferred from the SADCP data shows that this contrast is also observed at depth with slightly larger velocities in the western part of our study area (Fig.2b). There the bathymetry ranges typically from 4000m up to a few hundred meters locally with significant topographic slopes, which is consistent with the higher velocity signal; by comparison, in the East the bathymetry is almost flat with $\sim 5000m$ depth. More insights on turbulence are given with vertical sections of $\epsilon$ and $K_z$ in Figure 3a and b. The range of $\epsilon$ values covers 3 orders of magnitude, $\sim [10^{-10}, 10^{-7}]Wkg^{-1}$ below the mixed layer **(magenta curve in Fig.3a)** down to 500m depth. $\epsilon$ presents a typical patchy pattern with spots of intense turbulence with values up to $\sim 10^{-8}Wkg^{-1}$ down to 500m. **These events are localized over a 10-m vertical scale except at LD-A around 165E longitude where a 200m layer of enhanced $\epsilon$ is observed.** Most of these events are observed in the West , West of 170W, and the few vertically localized large $\epsilon$ observed East of 170W are no deeper than $200m$ **depth** (Fig.3a). **Statistics of $\epsilon$ within a** $100m - 500m$ **depth interval are given for each region in Table 2. The contrast between the mean** $100m - 500m$ **depth averaged $\epsilon$ in each of the two regions is of a factor of 3 (Table 2). The contrast is larger for the standard deviation, of a factor of 10, which points out the larger intermittency of turbulent events west of 170W.** The $K_z$ pattern presents a similar contrast between the western and eastern parts **with mean** $100m - 500m$ **depth averaged** $K_z$ **between the two regions differing by a factor of 2 (Table 2). More importantly, the standard deviation of $K_z$ is by**

**far larger west of** $170W$**, by a factor of 15 (Table 2)**.

## 3.2  Shear instability

To gain further insights on the origin of this contrast in turbulence, the possible occurrence of shear instability is addressed with buoyancy frequency, $N$, shear, $S$ and $N^2 - S^2$ sections displayed in Figure 4. The stratification is strong in the first 100m with a pycnocline that is generally well marked except westward of 165E and around 172W longitudes **and a less stratified surface layer above 50m east of 170W**. The shear is significantly higher west of 170W with high values over the **500m depth layer** of the measurements in the West **except for a** $165E - 170E$ **region** (Fig.4a and b). **The likeliness of shear instability was estimated from** $N^2 - S^2$**, displayed in Figure 4c. The upper 100m surface layer is the most stable with the highest** $N^2 - S^2$ **values as a result of both strong stratification and low shear.** Shear instability is more likely to occur west of 170W **below the 100m surface layer of strong stratification and where the shear is large: the percentage of data points that verifies the criterion for shear instability is ten times larger west of 170W than east of 170W (Table 2). The fact that** most of the subcritical $Ri$, $N^2 - S^2 < 0$, are observed **west of 170W is consistent with** the enhanced dissipation observed there (Fig.3 and 4c).

## 3.3  Tidal and atmospheric forcings for the internal wave field

The possible impact of internal waves was estimated indirectly through the two main energy sources for these waves, namely tidal forcing and wind power input (Fig.5 and 6).

The depth-integrated tidal generation force is displayed in Figure 5 for the $K1$ and $M2$ constituents. There is a strong similarity between the two constituents with a generation that is favoured in the western part which is shallower and with stronger topographic gradients than the eastern part of our study area. The most western region is characterized by numerous spots of generation with a depth-integrated generation force of $10^3 m^2 s^{-2}$ while eastward of 170E longitude there is only one main generation site around 180**E** longitude (Fig.5a). This spatial distribution of the internal tide forcing might suggest a similar contrast in the internal tide induced dissipation since the high modes responsible of turbulence are expected to dissipate within a few tens of kms of the generation site (e.g., St Laurent et al., 2002).

Maps of wind power input on inertial motions, also referred to as inertial flux, were computed using the spectral method described by Alford (2003). The wind stress data were inferred from WRF numerical simulations (Klemp et al., 2007; Skamarock et al., 2005) and the seasonal climatology was used for the mixed layer depth. The power input into inertial motions gives insights on the generation of baroclinic near-inertial waves (niw) at the base of the mixed layer through inertial pumping (e.g., Gill, 1984). The maps reveal a strong longitudinal contrast in inertial flux until mid March (Fig.6a-e). The strongest wind power input was observed in the western part of our study area. This is consistent with the climatology of storms and cyclones in the area that are typically formed in the SW tropical Pacific (e.g. Diamond et al., 2013). At the beginning of the cruise the largest power input was localized SW of the cruise stations (Fig.6a). Later a major event was observed (Fig.6d) during the passage

of a tropical cyclone over the area while the RV l'Atalante was sampling to the East. Eventually by the end of the cruise the inertial flux was small over the OUTPACE region with one spot of weak inertial flux observed in the East (Fig.6f-g). These maps suggest that energetic niw are likely to be generated in the western part of our study area prior to the cruise and until mid-March (Fig.6a-b). The first event of large inertial flux prior to the cruise may be particularly insightfull since it is likely to lead to the generation of equatorward propagation niw within the OUTPACE sampling area, a scenario which is consistent with large $\epsilon$ values there (Fig.1).

## 4 Possible impact of internal waves: focus on long duration stations

Three long duration stations were sampled each over three inertial periods, LD-A in the western part of the transect and LD-B and LD-C in the eastern part of the transect (see Table 1 and Fig.1). Turbulence at LD-A is by far the largest down to **5**00m depth with contrasted mean $\epsilon$ and $K_z$ between LD-A on one hand and LD-B and LD-C on the other hand (Fig.7a and b) **by** a factor of 7 and 5 for the averaged, within $[100m, 500m]$, $\epsilon$ and $K_z$ respectively **(Table 3)**. Possible occurrence of shear instability is examined by comparing mean profiles of shear square, $S^2$, and $N^2$ (Fig.7c). While the mean stratification is fairly close at the three stations the shear square is larger at LD-A compared to LD-B and LD-C within a factor of 10 within $[50m, 250m]$ (Fig.7c). Furthermore within $[100m, 200m]$ $S^2$ is larger than $N^2$ at LD-A pointing out possible shear instability. This depth range coincides with local $\epsilon$ maxima thus reinforcing the shear instability hypothesis.

We next focused on the characterization of the internal wave field that may reinforce the vertical shear and contribute to the onset of turbulence. Currents magnitude at LD-A is the largest (Fig.8a and b), of the order of $0.4ms^{-1}$. Detailed insights from the 150kHz SADCP reveal a wavy pattern with two frequencies clearly identified (Fig.8a): strong upward propagating bands close to the inertial period, of about 1.5 day, and the semi-diurnal period, which manifests itself through semi-diurnal heaving of upward propagation niw bands. The former is observed over the first two hundred meters only while the latter is observed down to $\sim 800m$ depth (Fig.8a and b). The weaker currents at LD-B and LD-C are comparable with **a** maximum amplitude of $0.2ms^{-1}$ (Fig8c-f). Periodic motions are also evidenced with inertial oscillations in the first few meters (Fig8c) and a combination of near-inertial and tidal periods at depth. Noticeably an upward phase propagation of niw can be inferred at LD-B from the 38kHz SADCP data (Fig8d). At LD-C the semi-diurnal tidal signal dominates (Fig.8f). The dominance of niw at LD-A is consistent with the highest wind power input inertial motions at LD-A (Fig.6a) compared to LD-B and LD-C (Fig.6e). Instead the contrast observed between the semi-diurnal depth-integrated generating force at LD-A compared to that at LD-B and LD-C (Fig.5b) is not evidenced on the semi-diurnal currents (Fig.8). This is well evidenced below $\sim 300m$ depth where the semi-diurnal tidal signal dominates at all stations (Fig.8b, d and f). This difference might result from localized generation areas of small scales that are not predicted by the estimate performed with low resolution fields (tidal model and bathymetry) or from low modes with long range propagation.

Figure 9 summarizes the main characteristics of the three long duration stations, **with vertical profiles of time-averaged $\epsilon$ and kinetic energy for the sub-inertial flow, the inertial frequency band and the semi-diurnal tidal constituent, $M2$. Kinetic energies were inferred from the frequency spectra computed over three inertial periods. Depth-averaged values**

of $\epsilon$, $K_z$ as well as kinetic energy and shear within the different frequency bands are also given in Table 3. The average was computed over two depth intervals: the first one is $100m - 500m$ consistently with that chosen in Table 2 while the second one is $50m - 250m$ corresponding to the depth interval of maximum niw energy at LD-A. The enhanced $\epsilon$ at LD-A is coincident with an energetic niw (Fig.9a). The contrast with LD-B and LD-C is striking within the depth interval $50m - 250m$ with an averaged $\epsilon$ within $50m - 250m$ about 8 times larger than at LD-B and a near-intertial kinetic energy that differs by almost an order of magnitude (Table 3). The significant decrease in $\epsilon$, coincident with a sharp shutdown of the near-inertial baroclinic signal around 250m, shows the main effect of niw on dissipation. This transition is associated with a strong variation in the subinertial flow that suggests a wave mean flow interaction (e.g. critical level; Soares et al., 2015). LD-B and LD-C present strong similarities in $\epsilon$ and niw and M2 kinetic energies with $100m - 500m$ depth averaged values that are close (Table 3). The local maxima in near-inertial kinetic energy may evidence niw beams around 150m, 450m and 650m at LD-B and 200m, 350m, 400m and 650m at LD-C (Fig.9b and c). The semi-diurnal kinetic energy presents an interesting contrast between LD-B and LD-C: while it is larger at LD-B in the first 500m and smaller below the opposite is observed at LD-C with maximum semi-diurnal energy below 500m depth, also suggesting a beam structure. The subinertial flow is the weakest at LD-C (Fig.9c) by a factor of 2 compared to LD-B and by a factor of 8 compared to LD-A for the $100m - 500m$ depth-averaged value of the kinetic energy. Moreover the depth-averaged low-frequency shear is below the noise level at LD-B and LD-C (Table 3), both features suggesting a weak influence on internal wave propagation. The subinertial flow is by far the largest at LD-A down to $\sim 250m$ (Fig.9a). The contrast in turbulence between the three stations is mostly confined in the upper few hundred meters as a result of an energetic niw and its interaction with the strongly sheared subinertial flow (Table 3). Deeper, variations in $\epsilon$ and kinetic energies are much weaker and of the same order of magnitude at the three stations (Fig.9).

## 5    Impact of turbulence on biogeochemical fluxes: spatial pattern and intermittency

### 5.1    overview along the OUTPACE section

The distribution of chlorophyll concentrations along the OUTPACE transect is typical of a transition from an oligotrophic area in the MA toward an ultraoligotrophic area in the SPSG (e.g., Moutin et al., 2017) with a deepening of the deep chlorophyll maximum, DCM, from $\sim 60m$ to $\sim 160m$ and an increase of the euphotic zone depth (Fig.10a). There is one noticeable exception to this trend with a near surface chlorophyll concentration maximum at $\sim 35m$ depth, at LD-B. de Verneil et al. (2017), who focused on the characterization of this anomalous phytoplankton bloom event, explained its occurrence by the main impact of mesoscale advection and an island effect. The deepening of the DCM results from that of the nitracline and from the $NO_3$ depletion in the first 200m, an evolution consistent with the increasing oligotrophy to the East (Fig.10a and c). Moreover east of 172W the nitracline is most often deeper than the base of the euphotic layer which reinforces the oligotrophy. The turbulent diffusive nitrate flux displays the same longitudinal trend (Fig.10c) with a depth-averaged value in the photic layer that differs by almost a factor of 4 west of 170W and east of 170W (Table 4). The standard deviation is as well by far larger in the west, by a factor of 15. Whether these flux variations are driven by $K_z$ variations or $c_z$

variations was examined within a 100-m depth interval below the nitracline all along the section. These large variations of the turbulent diffusive nitrate flux at small scales (Fig.10c) mostly result from $K_z$ variations with a weaker contribution of variations in the vertical gradient of nitrate concentration with a typical ratio of 3 between the two (Table 5). The latter contribution tends to weaken slightly the eastward decrease of the flux due to decreasing $K_z$. The variation of the nitrate diffusive flux was also examined at LD stations only in order to focus on the impact of the temporal intermittency of turbulent events at each LD stations where a large number of VMP profiles were collected. The flux variation resulting from $K_z$ variations, $-\Delta K_z c_z$, and those resulting from $c_z$ variations, $-K_z \Delta c_z$, were compared to the averaged flux over the three LD stations. The $-\Delta K_z c_z$ always dominates and is by far the largest at LD-A with a relative value of $105\%$ to be compared with the $-50\%$ and $-55\%$ relative values at LD-B and LD-C (Table 5). This shows the major impact of the turbulence intermittency induced by the niw event at LD-A.

Phosphate turbulent diffusive flux is in average smaller than the nitrate turbulent diffusive flux but its relative impact may be rescaled **by a factor of** $16 : 1$ **corresponding to the Redfield N:P ratio**. Hence, for visual comparison between the two fluxes, the scale for the phosphate turbulent diffusive fluxes differs from that of the nitrate turbulent diffusive flux within the Redfield ratio in Figures 10 and followings. The phosphate turbulent diffusive flux displays the similar longitudinal gradient **in the first 200m** but **presents an opposite trend in the photic layer with a depth-averaged value higher by a factor of 1.5 east of 170W (Table 5). The concentration of phosphate is not as strongly limiting as that of nitrate in the photic layer with significant turbulent diffusive flux, in a few spots, especially around 170 and 190 longitudes (Fig.10d). Consistently the same trend is obtained for the phosphate concentration in the photic layer with an eastward increase in the photic layer as opposed to the nitrate concentration that tends to zero at the eastern stations (Fig.10c and d). The absolute value of the photic layer depth averaged value of the flux, east of 170W, is of** $4.01 \mu mol m^{-2} d^{-1}$. **Considering a N:P Redfield ratio of 16, the phosphate turbulent diffusive flux is significant compared to the** $3.15 \mu mol m^{-2} d^{-1}$ **value for the nitrate turbulent diffusive flux (Fig.10e and f, Table 4).** This striking difference in phosphate and nitrate turbulent diffusive fluxes within the photic layer may play an important role on the development of micro-organisms as discussed later.

## 5.2 Focus on LD stations

Turbulent diffusive fluxes of nitrate and phosphate were further analyzed at long duration stations (Fig.11). The time-depth evolution of $K_z$ underlines the very large values encountered at the most turbulent station, LD-A, in contrast with values at LD-B and LD-C that show the occurrence of a few spots of intense mixing in a more quiescent background with $K_z \sim 3 10^{-6} m^2 s^{-1}$ (Fig.11a, d and g). The largest nitrate turbulent diffusive fluxes occur at LD-A (Fig.11b) while the smallest values are observed **at LD-B and** LD-C (Fig11f and i). **The averaged values in the photic layer shows that it is only at LD-A that there is a small input of nitrate through diffusion with an average flux of** $8.41 \mu mol m^{-2} d^{-1}$ **(Table 4).**

**In contrast** phosphate turbulent diffusive fluxes are significant well above the **euphotic zone depth at all LD stations** (Fig.11b, e and h) **which may have an impact on primary production whereas the nitrate input by turbulent diffusion is negligible (explanation below). Depth-averaged values in the photic layer are even comparatively larger than that of the nitrate**

turbulent diffusive fluxes at LD-A if one applies the P/N=1/16 Redfield ratio (Table 4). Various spots of large phosphate turbulent diffusive fluxes are also evidenced in the first $\sim$20-80-m that can be correlated with events of intense turbulence (Fig.11b, e and d). At LD-C the only event of significant phosphate turbulent diffusive flux results from a strong tu rbulent event (Fig.11h).

### 5.2.1 Nitrate input at the top of the nitracline

The input of nitrate through turbulent diffusion **was first examined at the top of the nitracline, within a $\sim 4m$ tickness layer, with histograms of the nitrate turbulent diffusive flux (Fig.12). There is a strong contrast between LD-A and the eastern stations in the shape of the histograms: an atypical shape of the histogram is obtained at LD-A with a large standard**

**deviation while the distribution of the nitrate turbulent diffusive flux is fairly similar at LD-B and LD-C with one main peak (Fig.12d and f). The atypical shape obtained at LD-A results from the large intermittency of $K_z$ as illustrated in Figure 12a.** The distribution of $K_z$ values covers 2 orders of magnitude and presents a bimodal distribution. **The moderate $K_z$ values are associated with the 'background state' while the large values are** associated with intense turbulent events related to the near-inertial baroclinic wave (e.g. Fig.8a). **The mean value of the nitrate turbulent diffusive flux within a**

**$\sim 4m$ layer starting from the top of the nitracline, is smaller at LD-B compared to LD-A and LD-C: $24.1 \mu mol m^{-2} d^{-1}$ and $18.9 \mu mol m^{-2} d^{-1}$ at LD-A and LD-C to be compared with the mean value of $6.0 \mu mol m^{-2} d^{-1}$ at LD-B. This contrast may appear surprising, with a comparable mean value at LD-A and LD-C, but this results from the occurrence of a few turbulent events at LD-C where the nitracline is the deepest. Nevertheless the main point regarding the impact of the turbulent input of nitrate at the top of the nitracline on new primary production is whether or not the top**

**of the nitracline falls within the photic layer. This point is addressed in the following with histograms in the photic layer.**

### 5.2.2 New primary production sustained by phosphate turbulent diffusive fluxes at the western stations?

Figure 13 summarizes the contrast between long duration stations in the **photic** layer with histograms of $K_z$, phosphate and nitrate turbulent diffusive fluxes. **The euphotic zone depth (EZD) was immediately determined on board from the**

**photosynthetically available radiation (PAR) at depth compared to the sea surface PAR(0+), and used to determine the upper water sampling depths corresponding to 75, 54, 36, 19, 10, 3, 1 (EZD), 0.3, and 0.1% of PAR(0+) (e.g. Herbland and Voituriez , 1977; Moutin and Prieur , 2012). The euphotic zone depth varies within $[55m - 120m]$ with a mean value of $70m$ at LD-A, $55m$ at LD-B and $120m$ at LD-C.** As in the previous figures the scales for the nitrate and phosphate turbulent diffusive fluxes match with the Redfield ratio for visual comparison of the relative impact of each of these fluxes on

micro-organisms. Significant nitrate turbulent diffusive flux is observed at LD-A as opposed to the LD-B and LD-C as a result of shallower nitracline that falls within the photic layer at LD-A (Fig.13c, f and i). At LD-B and LD-C where the nitracline is below the euphotic zone depth the nitrate turbulent diffusive flux is zero (Fig.13f and i). The station average of the phosphate turbulent diffusive flux **is of the same order of magnitude at LD-A and LD-B and smaller by a factor of 10 at LD-C**

(Fig.13b, e and h; Table 4). These significant values of the phosphate turbulent diffusive flux observed at LD-A and LD-B suggest an impact on micro-organisms (Fig.13e). Indeed, the presence of nitrogen fixers (micro-organisms able to use the atmospheric $N_2$; Dupouy et al., 2000) and high rates of N input by $N_2$ fixation were already noticed in the Western Tropical South Pacific (Moutin et al., 2008) and confirmed in the whole area (Bonnet et al. , 2008). In contrast the smallest values observed at LD-C (Fig.13h) are consistent with low nitrogen fixation rates measured there (Bonnet et al., 2017) or in the whole South Pacific gyre (Moutin et al., 2008), probably because of iron depletion (Bonnet et al., 2017; Moutin et al., 2008; Blain et al., 2007; Guieu et al., 2018). The lack of iron for $N_2$ fixers may explain their lower presence in the gyre and consequently the relatively higher phosphate concentrations measured there in the upper layer. Phosphate is not depleted by the $N_2$ fixers even with relatively low turbulent diffusive fluxes of phosphate from below.

Figure 14 summarizes the main features of the turbulent diffusive fluxes with time-averaged vertical profiles. The double x-axis for the nitrate and phosphate turbulent diffusive fluxes is scaled within a Redfield ratio so that the fluxes are superimposed if they follow the Redfield proportion. At depth, $N0_3$ and $PO_4$ fluxes follow the Redfield proportion. Closer to the surface, $N0_3$ flux decreased: at LD-A there is a small input of nitrate in the photic layer while at the eastern stations the nitrate flux vanishes above the base of the euphotic zone and reached zero before $PO_4$ fluxes. These significant phosphate fluxes at shallower depths may potentially fueling nitrogen fixation. Significant $PO_4$ sources through turbulent diffusion that are likely to provide the required conditions for the growth of $N_2$ fixers in the Melanesian archipelago. Because phosphate availabilty likely sustain $N_2$ fixation and therefore N input by $N_2$ fixation in the WTSP, new primary production (Dugdale and Goering, 1967) might be sustained by new P (or new N including $N_2$ fixation) in the photic zone.

### 5.2.3 Turbulent diffusion and the oligo to ultraoligotrophic conditions encountered during the OUTPACE cruise

The decrease of nitrate turbulent diffusive flux eastward was found to be consistent with the increasing oligotrophy and the deepening of the nitracline (e.g., Moutin et al., 2018; Caffin et al., 2018). As a result the nitrate input into the photic layer through turbulent diffusion was found to provide only a subordinate contribution to the N budget with a $1 - 8\%$ contribution to the new N (Caffin et al., 2018). In the Melanesian Archipelago, the input of nitrate into the photic layer represents a small contribution during the stratified period (Caffin et al., 2018) as well as on an annual time scale with a $46 \mu mol m^{-2} d^{-1}$ input by turbulent diffusion to be compared with a $642 \mu mol m^{-2} d^{-1}$ input of N by $N_2$ fixation ((Table 6; Moutin et al., 2018).

In a $100 - m$ layer starting from the top of the nitracline, the mean nitrate turbulent diffusive flux varies by a factor of 3 between the western LD station LD-A with a flux of $\sim 45 \mu mol m^{-2} d^{-1}$ and the eastern LD-B and LD-C stations. The mean value obtained at LD-A is smaller than the average value obtained in the oligotrophic eastern Atlantic, $\sim 140 \mu mol m^{-2} d^{-1}$, by Lewis et al. (1986), suggesting an increased oligotrophy in the Pacific. It is typically one order of magnitude smaller than the values inferred further south, $\sim 30S$, by Stevens et al. (2012), where both larger Kz and nitrate vertical gradient are responsible for a larger nitrate turbulent diffusive flux. The comparison with Lewis et al. (1986) in the Atlantic ocean also highlights the ultra-oligotrophy of the Pacific ocean in the gyre compared to

**their counterpart in the Atlantic ocean with diffusive fluxes at least one order of magnitude smaller as shown at LD-B and LD-C. The low and deep nitrate turbulent diffusive fluxes may not explain the higher primary production and $N_2$ fixation rates observed in the upper 0-40 m (Moutin et al., 2018, their figures 6a and 6b).**

The input of phosphate in the photic layer was also addressed as a possible source for sustaining the development of $N_2$-fixing organisms. Phosphate turbulent diffusive fluxes mean values were significant in the photic layer with the exception of the most eastern station. In all cases, a few events of large fluxes driven by localized intense turbulent events were identified. The large variations in the turbulent diffusive fluxes resulting from the occurrence of strong turbulent events were thus underlined with a focus on long duration stations (see also Caffin et al., 2018). This rose the question of the estimate of the turbulent input of nitrate in the photic layer when establishing C, N and P budgets as well as the impact of turbulence intermittency on microorganisms (e.g., Liccardo et al., 2013).

## 6    Conclusions

Variations within a factor of 10 of the depth-integrated $\epsilon$ were observed along the OUTPACE transect. The largest $\epsilon$ observed in the West compare well with the few measurements performed by Fernández-Castro et al. (2014, 2015) in the area during the Malaspina expedition. The range of values is comparable with 80% of $\epsilon$ values within $[6 \times 10^{-10}; 10^{-8}]\,Wkg^{-1}$ in the first 300m below the mixed layer for the Malaspina expedition and 82% for the OUTPACE $\epsilon$ within $[30m, 300m]$ west of 180 longitude. Shear instability was evidenced as one main process responsible for turbulence which is a well-known mechanism in the strongly sheared Pacific equatorial currents (e.g. Richards et al., 2015; Smyth et al., 2013). Richards et al. (2012) also mentioned the modulation of the turbulence level over a 3 year period with different ENSO states in the western equatorial Pacific, with a maximum shear during La Niña events compared to El Niño events. How this turbulence cycle is relevant to the OUTPACE region would be an interesting point to address with possible higher turbulence level provided that the OUTPACE cruise took place during an El Niño event. Shear instability was found more likely to occur in the western part of our study area with most critical Ri encountered there. This basic analysis thus explained the contrast in dissipation observed along the transect. **Whether the onset of shear instability may be driven by the low frequency flow or the internal wave field could not be inferred from the short duration stations but insights on the impact of internal waves were given with estimates of the two main forcings for internal waves and the analysis of LD stations.** The main forcings of internal waves were found to vary significantly along the 19S longitudinal transect thus pointing out the possible impact of internal wave on the contrast in energy dissipation. The most striking factor was related to the atmospheric forcing with the occurrence of cyclones in the West leading to an energetic baroclinic near-inertial wave field. This internal wave component was characterized at the western long duration station, LD-A, as well as its impact on energy dissipation. These scenarios are typically encountered in tropical regions where baroclinic near-inertial waves are known to contribute to energy dissipation in the upper ocean (e.g., Cuypers et al., 2013; Soares and Richards, 2013). The process of dissipation is often constrained by the mean subinertial flow with ray focusing or critical levels depending on the spatial structure of the flow (e.g., Whitt and Thomas, 2013; Soares et al., 2015). These mechanisms were not addressed here but will be the focus of a future study using observations at the LD-A site.

**The context of the OUTPACE cruise with significant niw generated by a cyclone is very specific, of general interest for the studied region, where these meteorological phenomena are frequent at the end of the summer. It hides the more continuous influence of internal tides as a turbulence driver. Our measurements show only a slightly larger semi-diurnal kinetic energy in the West at LD-A compared to LD-B and LD-C but suggests a larger contrast in shear variance.**

The impact of turbulence on biogeochemical fluxes **and trophic gradients** was estimated based on nitrate and phosphate turbulent diffusive fluxes. **The nitrate input into the photic layer through turbulent diffusion was found to provide only a subordinate contribution to the N budget as a result of the eastward decrease of nitrate turbulent diffusive fluxes and the deepening of the nitracline.** New production is mainly supported by N$_2$ **fixation and located in the Western Tropical**
10   **South Pacific. The higher phosphate turbulent fluxes compared to nitrate fluxes provide an excess P relative to Redfield stoechiometry and a potential ecological niche for** $N_2$ **fixing organisms in the west. In addition to the lower iron availability in the east preventing N2 fixation to occur, the high iron availability in the west allow this process and the excess P provided to the photic zone sustain a higher new primary production explaining the western-eastern oligotrophy gradient.**

15   *Competing interests.*

*Disclaimer.*

*Acknowledgements.* This is a contribution of the OUTPACE (Oligotrophy from Ultra-oligotrophy PAcific Experiment) project (https://outpace.mio.univ-amu.fr) funded by the French research national agency (ANR-14-CE01-0007-01), the LEFE-CyBER program (CNRS-INSU), the GOPS program (IRD) and the CNES (BC T23, ZBC 45000048836). We warmly acknowledge the assistance of the crew of the French research
20   vessel l'Atalante, during the deployment of the VMP. We thank Olivier Desprez for his technical support and help during the cruise. The microstructure profiler was funded through the ANR OUTPACE project.

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

30  725.

## List of Figures

**List of Tables**

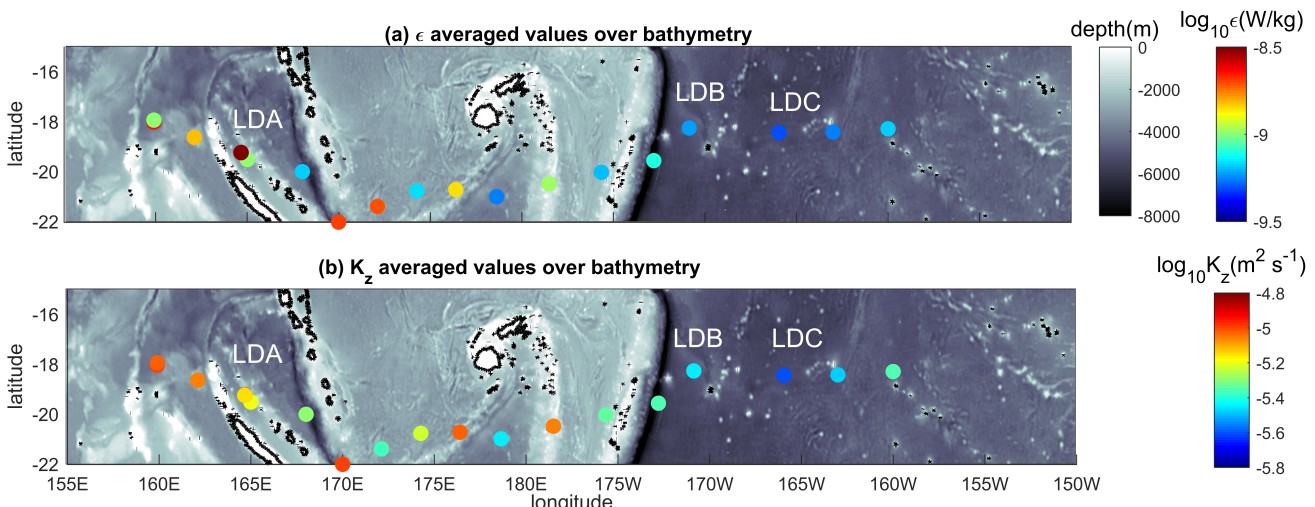

**Figure 1.** Dissipation rate of turbulent kinetic energy (a) and vertical diffusion coefficient (b) averaged below the mixed layer depth **(see Table 1 from Moutin et al, 2017)**, over $100m - 800m$ (log scale). Bathymetry is shown with gray color scale (ETOPO1 1 arc minute-Amante, C. and B.W. Eakins, 2009. ETOPO1 1 Arc-Minute Global Relief Model: Procedures, Data Sources and Analysis. NOAA Technical Memorandum NESDIS NGDC-24. National Geophysical Data Center, NOAA. doi:10.7289/V5C8276M). Time-averaged values **are displayed** at long duration stations, LD-A, LD-B and LD-C.

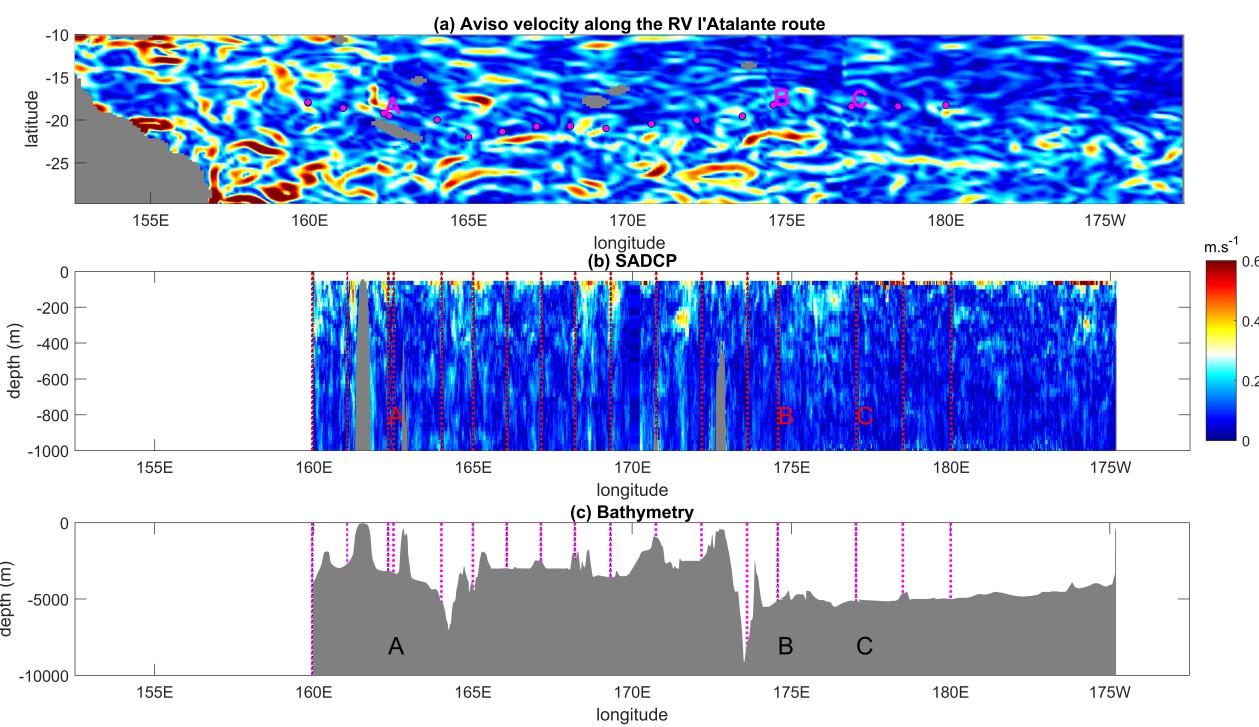

**Figure 2.** Surface geostrophic currents inferred from AVISO altimetric data, (a); longitude-depth section of 38kHz SADCP velocity modulus, (b); bathymetry along the RV l'Atalante route, (c); VMP stations are displayed with magenta circles in (a) and with vertical dashed red lines in (b) and magenta lines in (c).

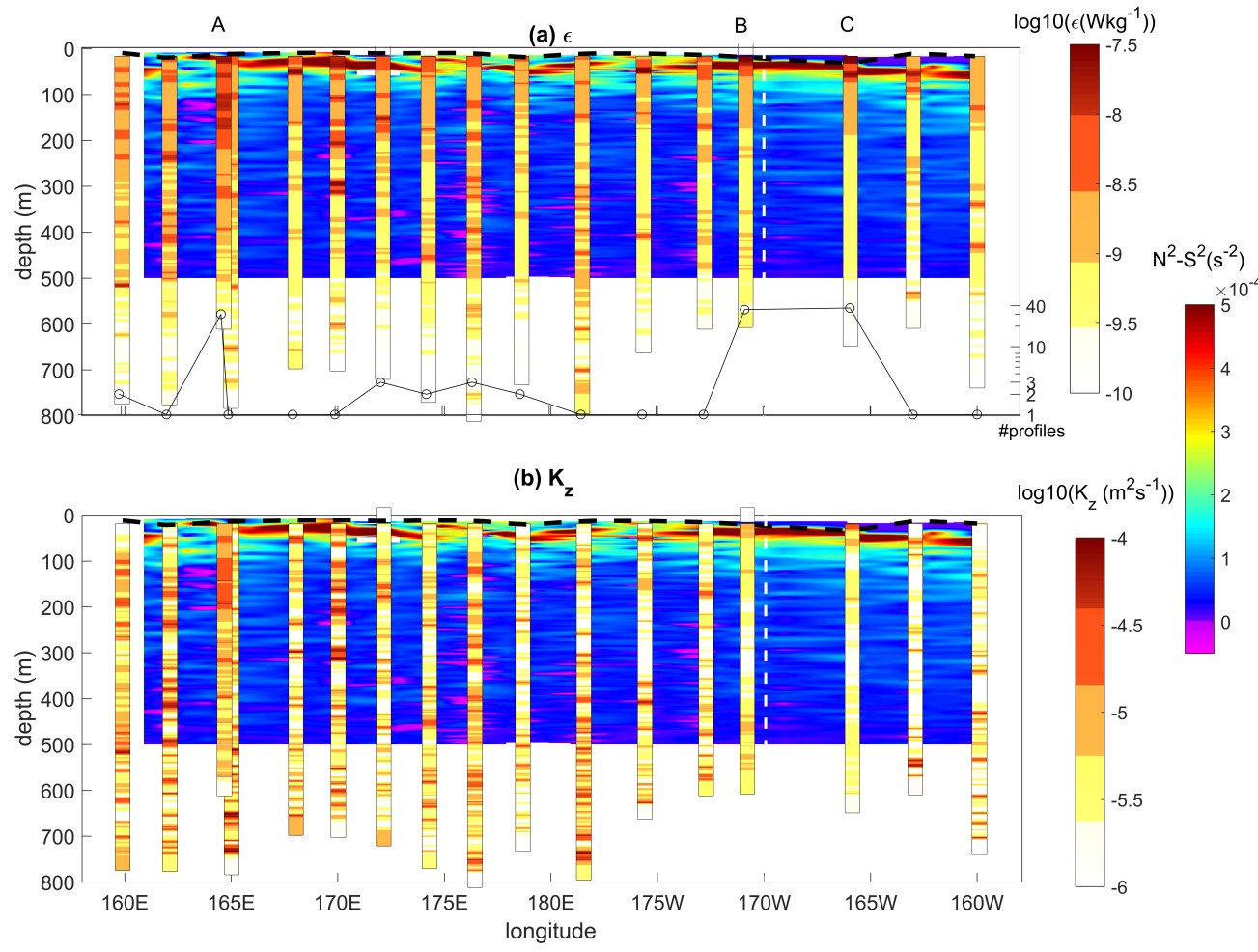

**Figure 3.** Log values of dissipation rate of turbulent kinetic energy ($W\,kg^{-1}$) (a) and vertical diffusion coefficient ($m^2\,s^{-1}$) (b) with $N^2 - S^2$ in background colorscale. **The number of profiles at each station is displayed with black circles and profiles have been time averaged at each station for better visualization. The mixed layer depth is plotted (black dashed curve) as well as the limit between the two regions (dashed white line).**

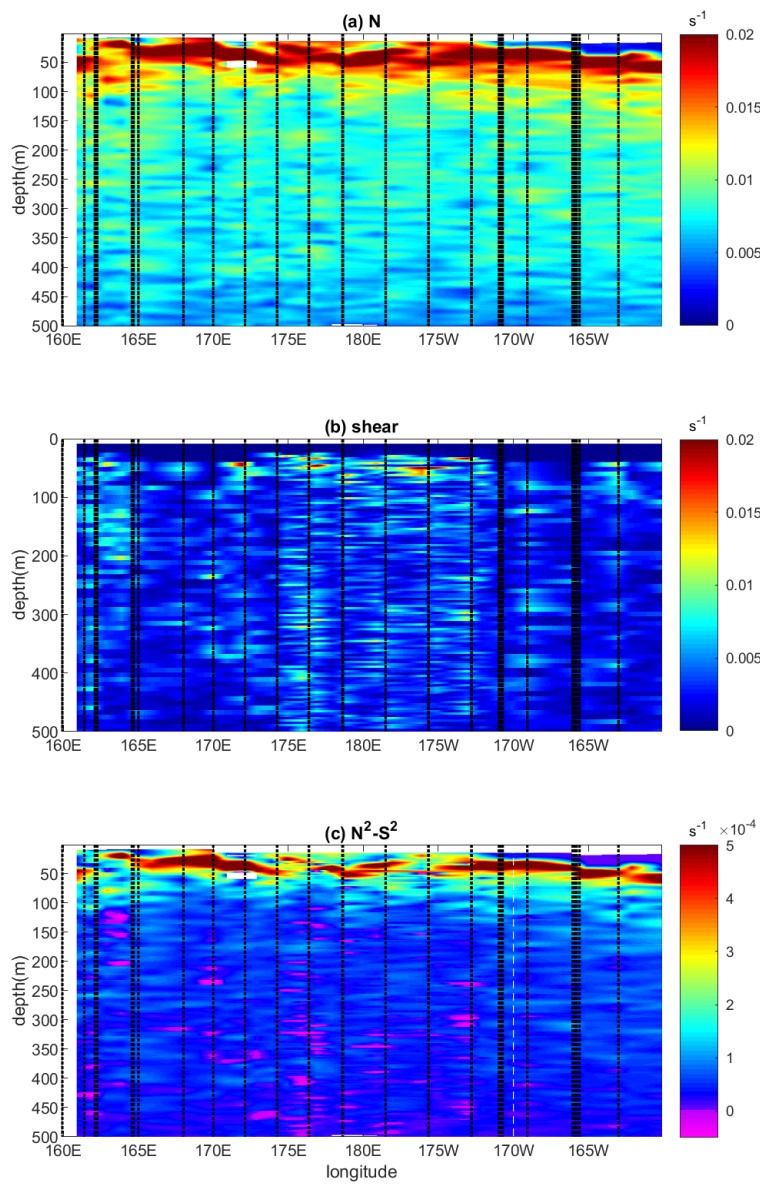

**Figure 4.** Buoyancy frequency, $N$, (a), shear, $S$, (b) and $N^2 - S^2$, (c) as a function of longitude and depth. The location of the CTD **and LADCP** profiles is displayed with black dashed lines. $S$ **is inferred from LADCP data and** $N$ **was vertically averaged over 8-m length vertical intervals for consistency with the 8-m bin of the LADCP data.**

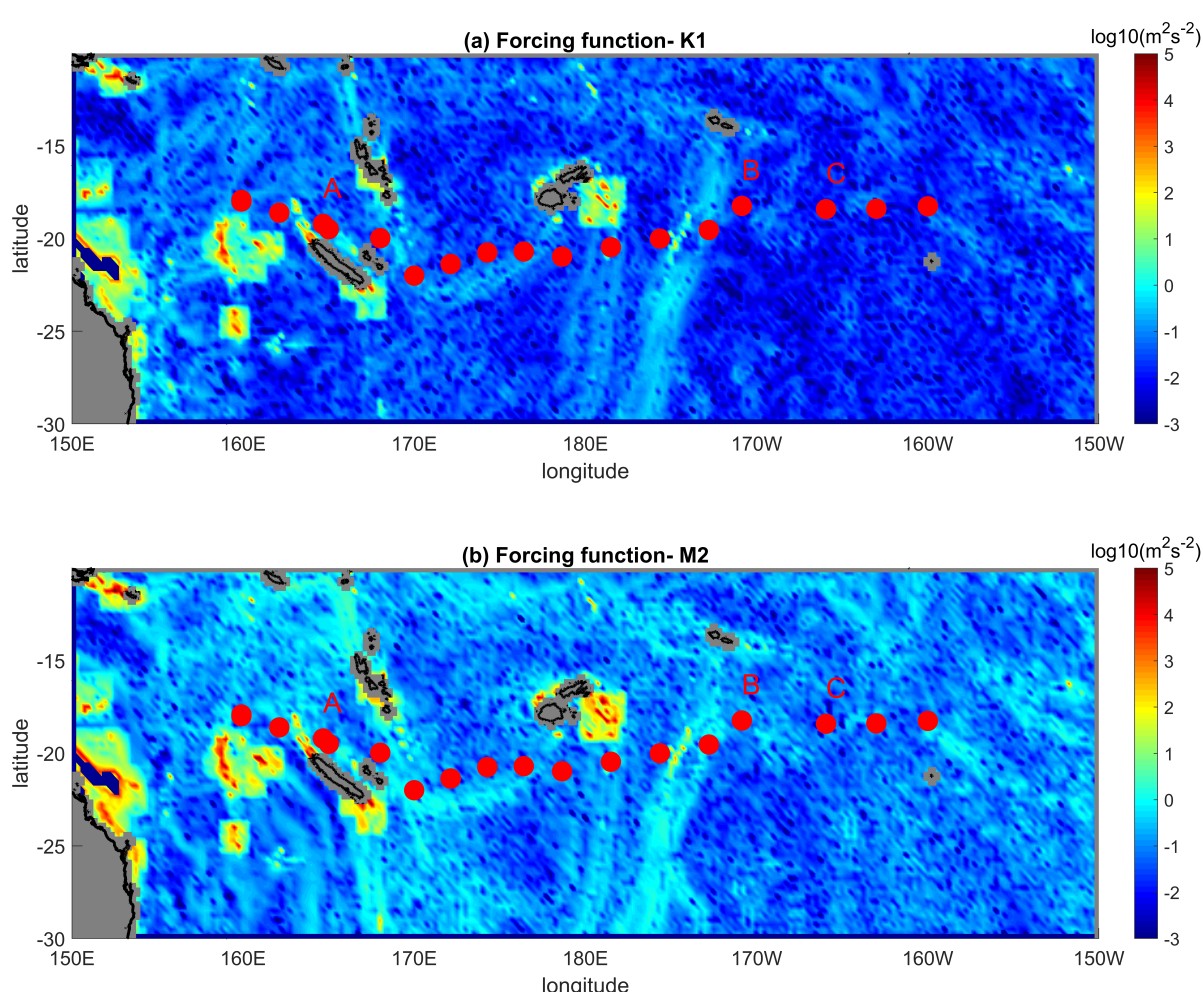

**Figure 5.** (a) Forcing function for K1 tidal constituent ($\log10(m^2 s^{-2})$); (b) same as in (a) but for M2 tidal constituent. The stations are shown with a red circle and the LD stations are indicated.

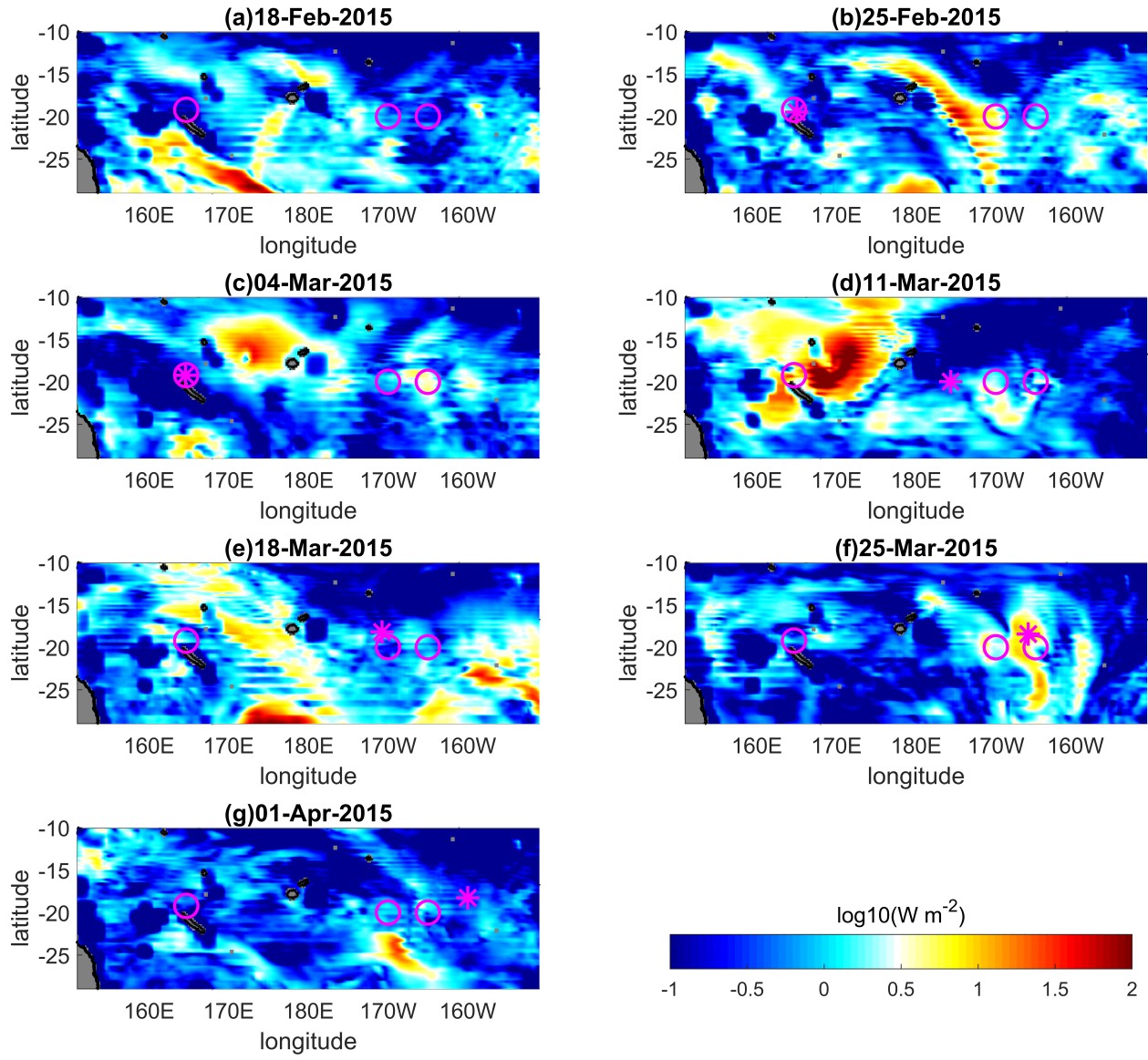

**Figure 6.** Maps of inertial energy flux (log10($W\,m^{-2}$)) every 7 days during the cruise. Long duration stations are shown with black circles, LD-A was hold during (b) and (c), LD-B during (e) and LD-C during (f). The ship position is displayed with a magenta star and the long duration stations with magenta circles.

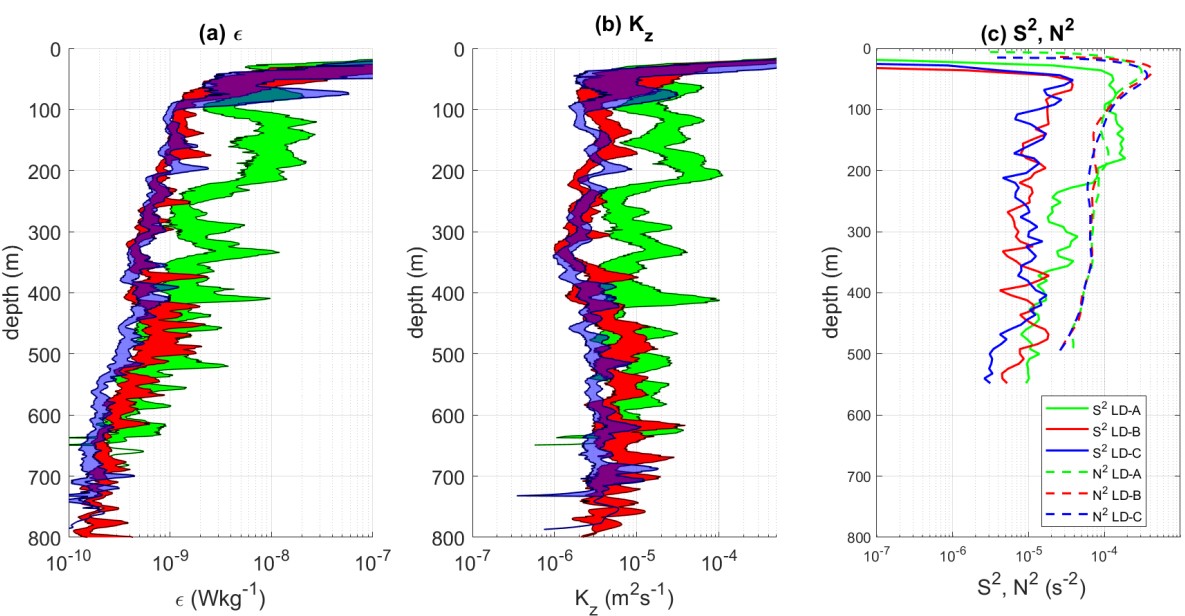

**Figure 7.** Mean profiles at long durations stations: $\epsilon$ in (a), $K_z$ in (b) and $S^2$ and $N^2$ in (c). $K_z$ was computed using $N$ from the VMP measurements while the profiles in (c) were inferred from the rosette mounted CTD and LADCP instruments. **The 95% confidence interval for $\epsilon$ and $K_z$ is displayed with a shading surface in (a) and (b).**

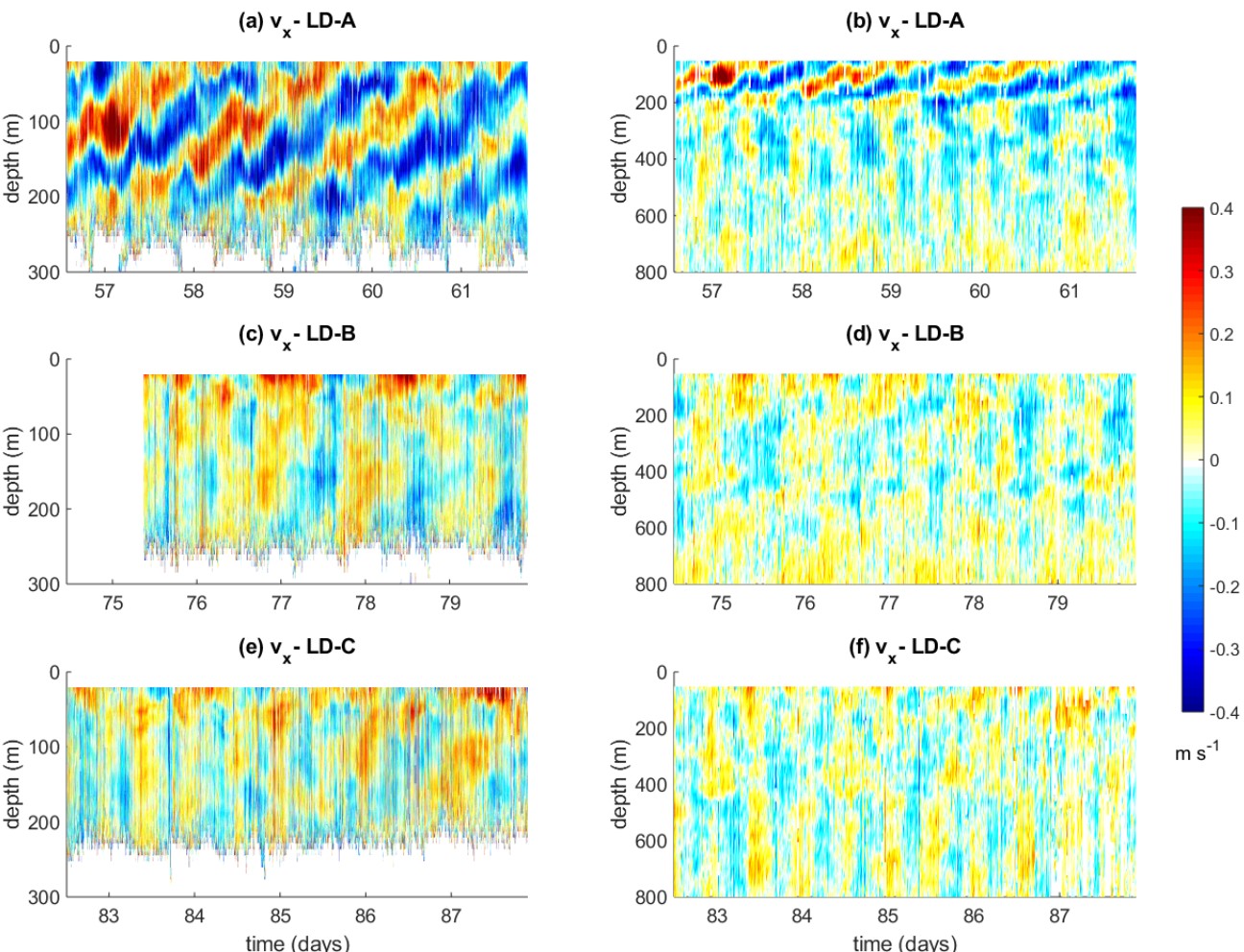

**Figure 8.** Zonal velocity ($m\,s^{-1}$) as a function of time and depth at the long duration stations, each row from 1 to 3 corresponds to LD-A, LD-B and LD-C respectively, in the first column ship ADCP data from the $150kHz$ instrument down to 300m are displayed and in the second column those from the $38kHz$ instrument down to 800m. Note that in (b) the 150kHz SADCP functionned only a few hours after the beginning of the station sampling.

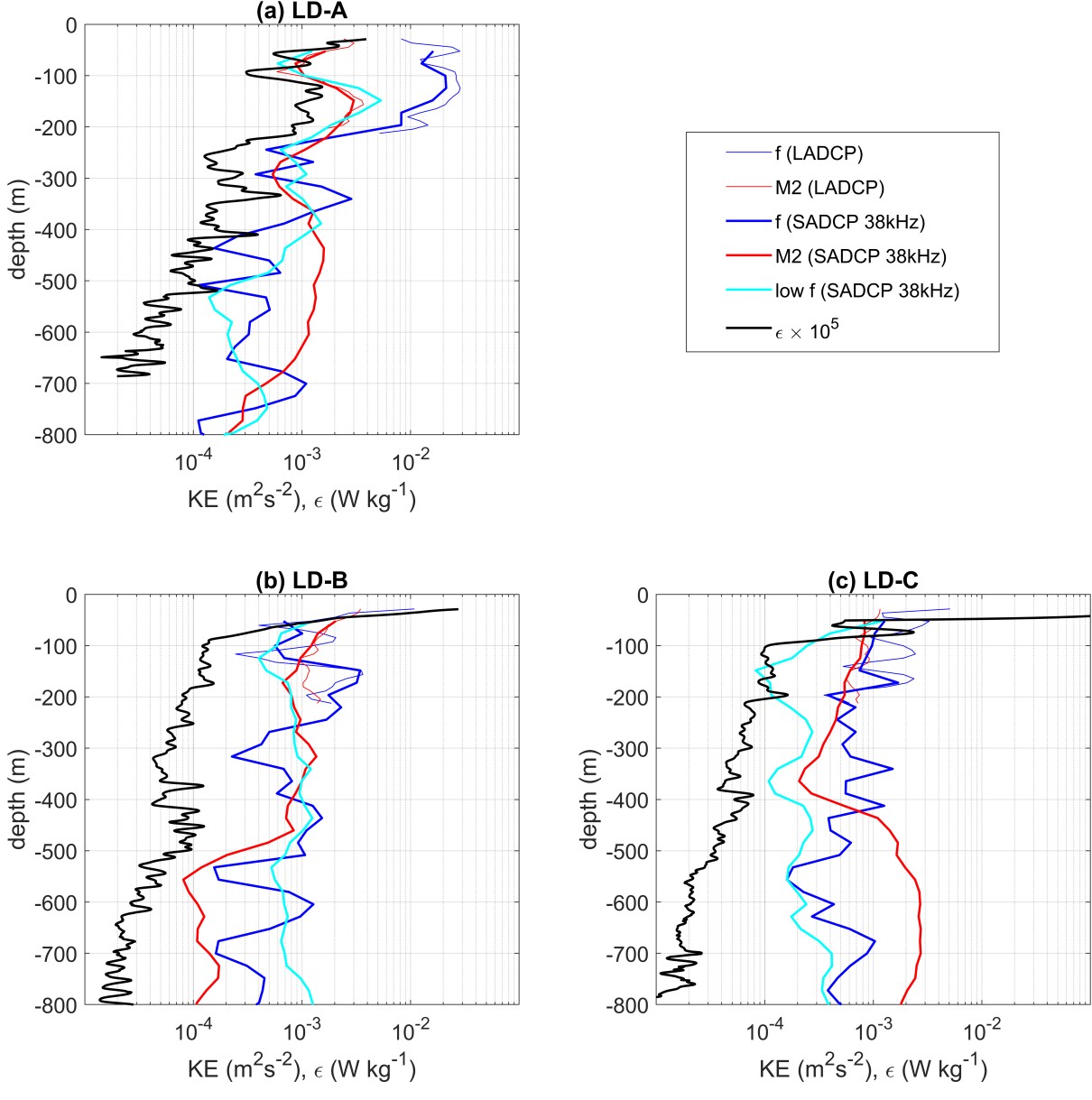

**Figure 9.** Time-mean profiles of the kinetic energy of the subinertial flow, the inertial and the semi-diurnal frequencies and $\epsilon$ at LD-A (a), LD-B (b), and LD-C (c). The kinetic energy was derived from the 38kHz SADCP data but also from the 150kHz SADCP data for the inertial and semi-diurnal kinetic energies (thin blue and red curves).

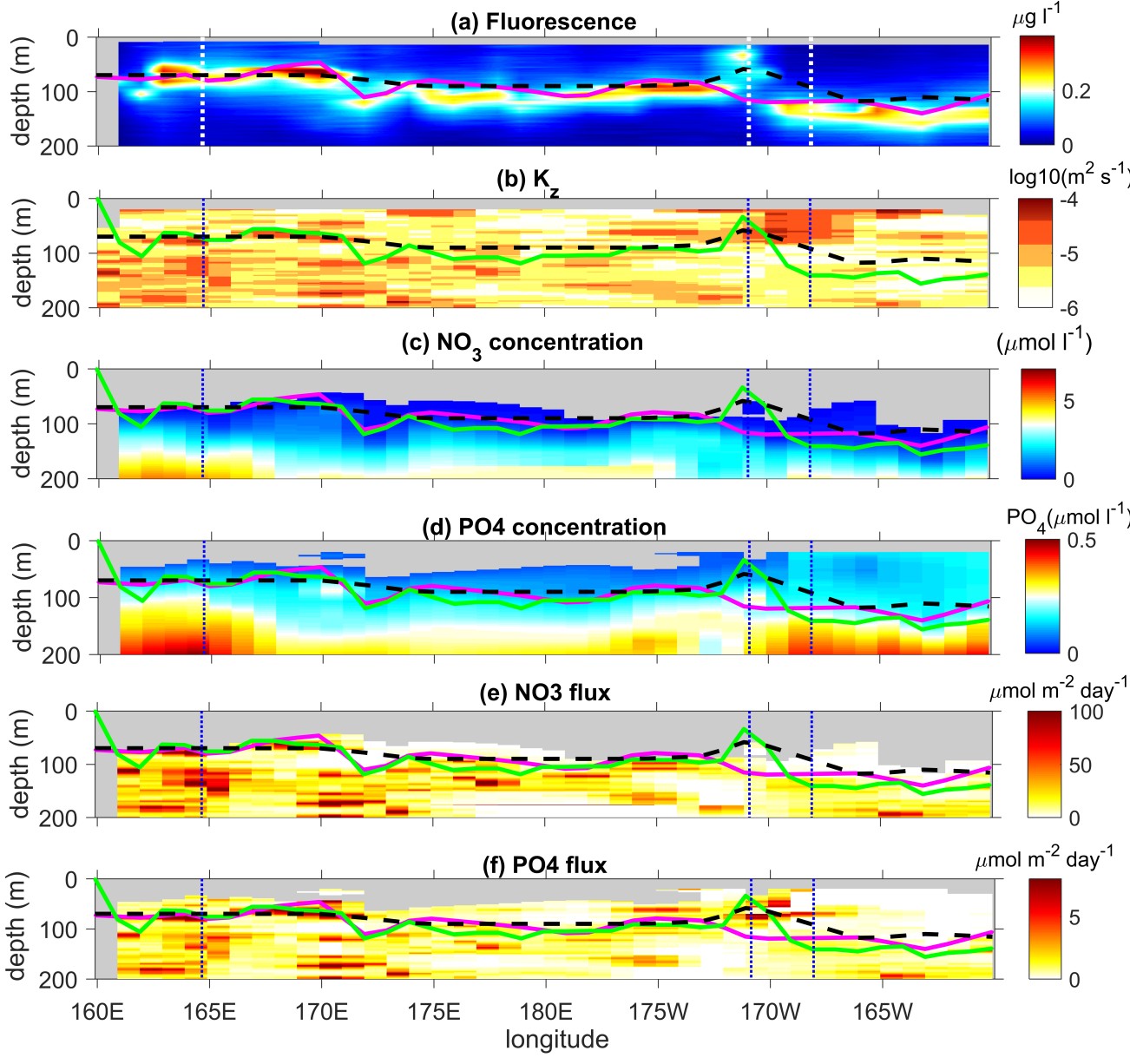

**Figure 10.** Longitude depth sections of **cholorophyll concentration**, (a), $K_z$, (b), **nitrate concentration, (c), phosphate concentration, (d),** nitrate turbulent diffusive flux (e) and phosphate turbulent diffusive flux (f). The top of the nitracline is shown with a magenta dotted line, the depth of maximum **chlorophyll concentration** with a green dotted line **and the euphotic zone depth with a dashed black line**. The location of the LD stations are shown with a vertical dashed line. **The scales of the nitrate and phosphate concentration and turbulent diffusive flux are set to allow a direct comparison of concentrations and fluxes against nutrient requirements for phytoplancton (Redfield proportion:** $N : P = 16 : 1$**).**

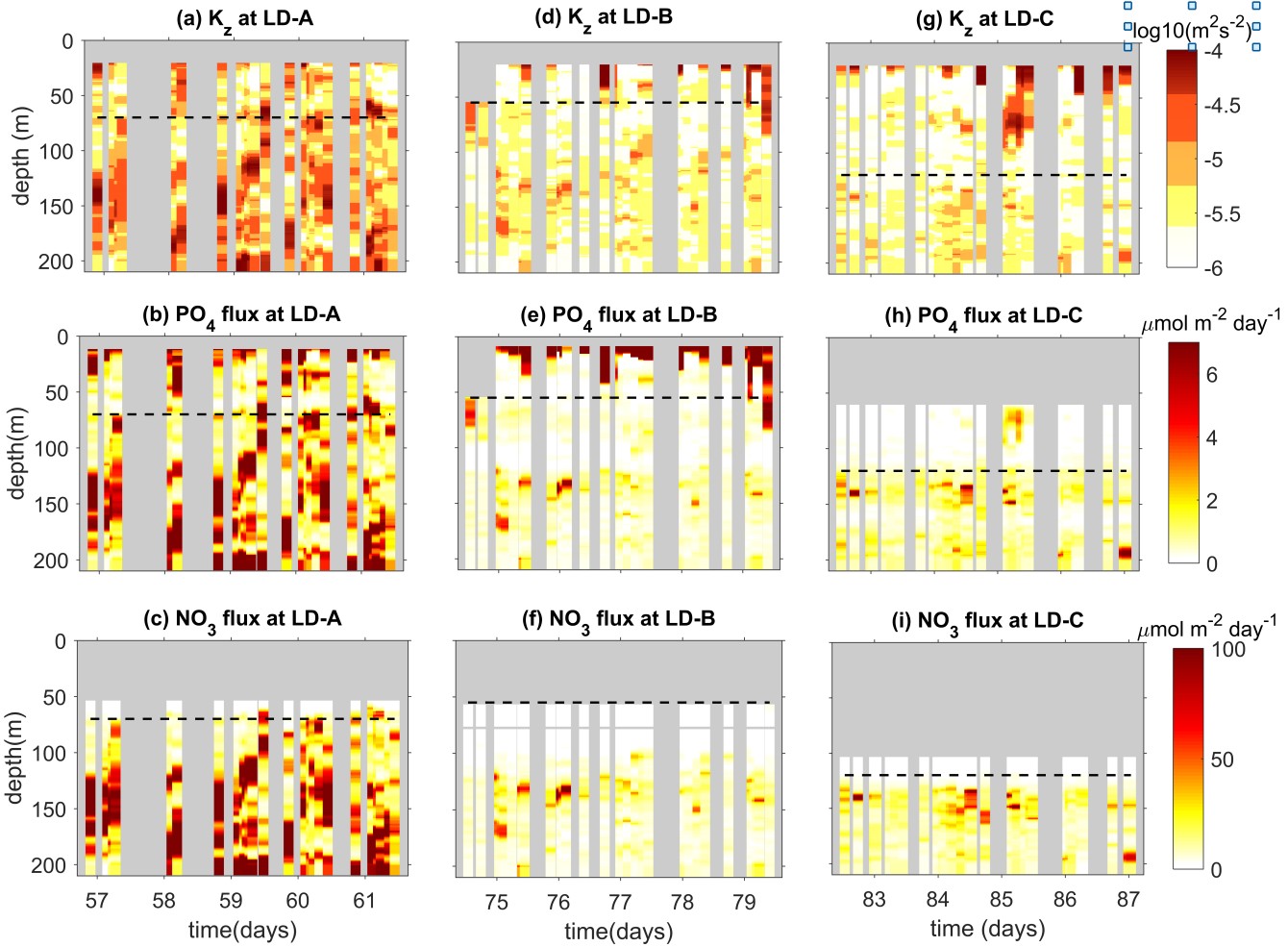

**Figure 11. Time** depth sections of $K_z$ (first **raw**), phosphate turbulent diffusive flux (second **raw**) and nitrate turbulent diffusive flux (third **raw**). The scales of the nitrate and phosphate diffusive flux are set to match with the typical Redfield ratio in the area ($N : P = 16 : 1$). **The mean euphotic zone depth is displayed with a black dashed line. The top of the nitracline is defined by the isopycnal $\rho_{NO_3}$ and falls at a depth of** $\sim 83.5m$ **at LD-A,** $\sim 111.2m$ **at LD-B and** $\sim 134.6m$ **at LD-C**

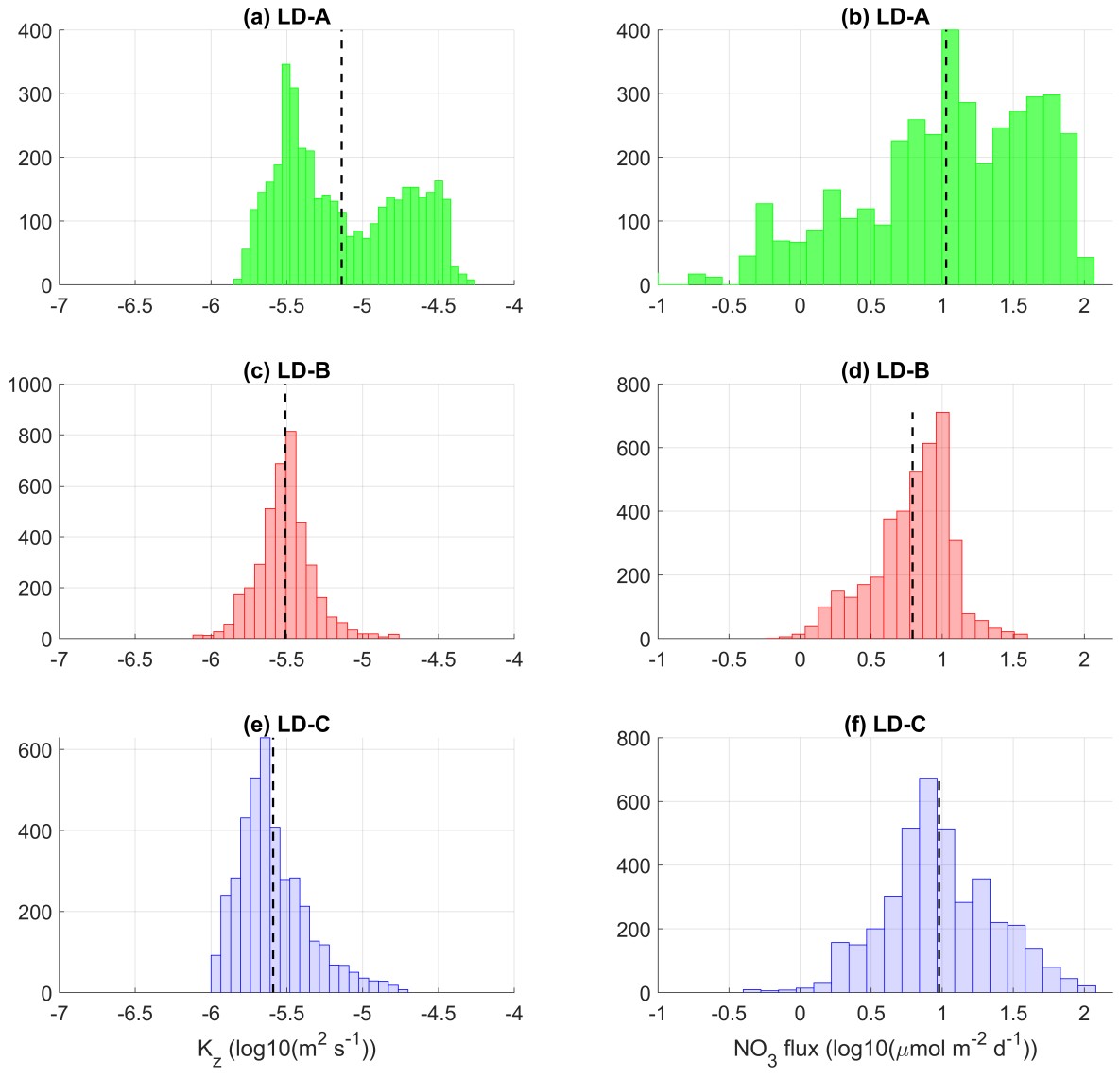

**Figure 12.** Histograms of $K_z$ (first column) and nitrate turbulent diffusive flux (second column) at long duration stations, LD-A, LD-B and LD-C around the top of the nitracline. The top of the nitracline is defined by the isopycnal, $\rho_{NO_3}$, with density values taken from Caffin et al. (2018), Table 4. **A density interval of upper bound equal to $\rho_{NO_3} + 3 \times 10^{-2}\, kg\, m^{-3}$, typically $3 - 4m$, was chosen. The mean value is shown in each subpanel with a dashed line.**

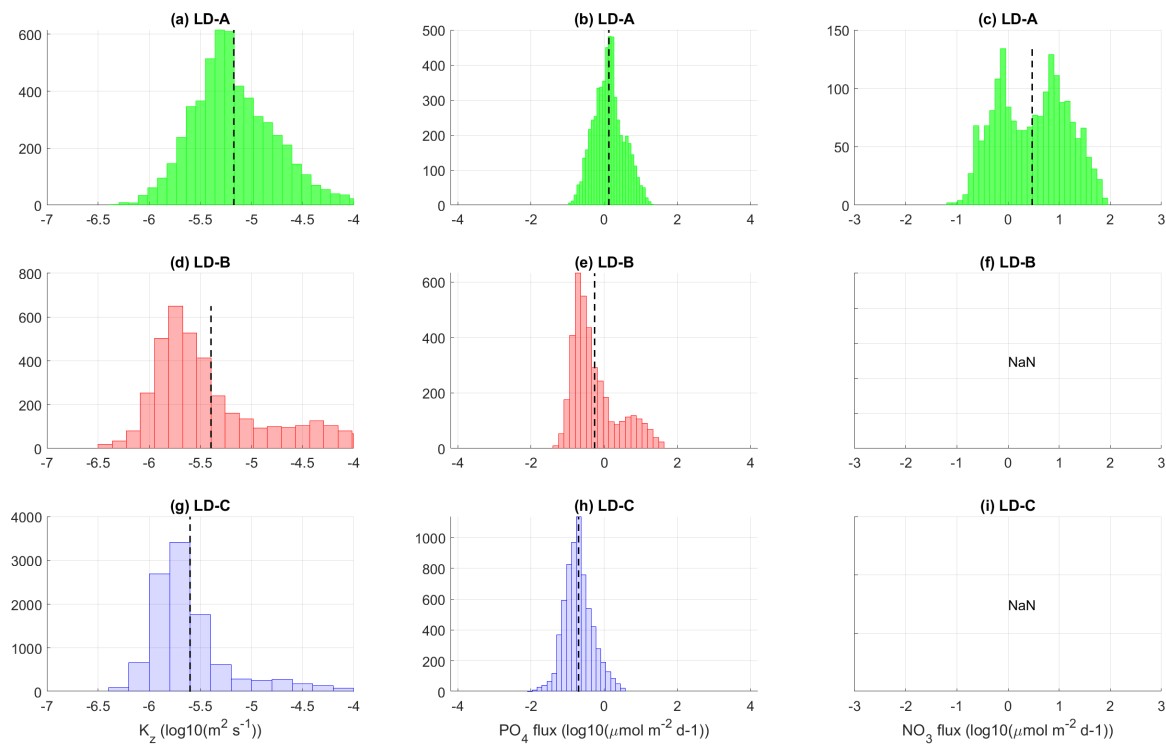

**Figure 13.** Histograms of $K_z$ (first column), phosphate and nitrate turbulent diffusive fluxes (second and third colums) at long duration stations, LD-A, LD-B and LD-C within the photic layer (down to 70m, 55m and 120m respectively). At LD-B and LD-C the nitrate turbulent diffusive flux is below the noise level within the considered depth interval, (f) and (i). **The mean value is displayed with a dashed black line.**

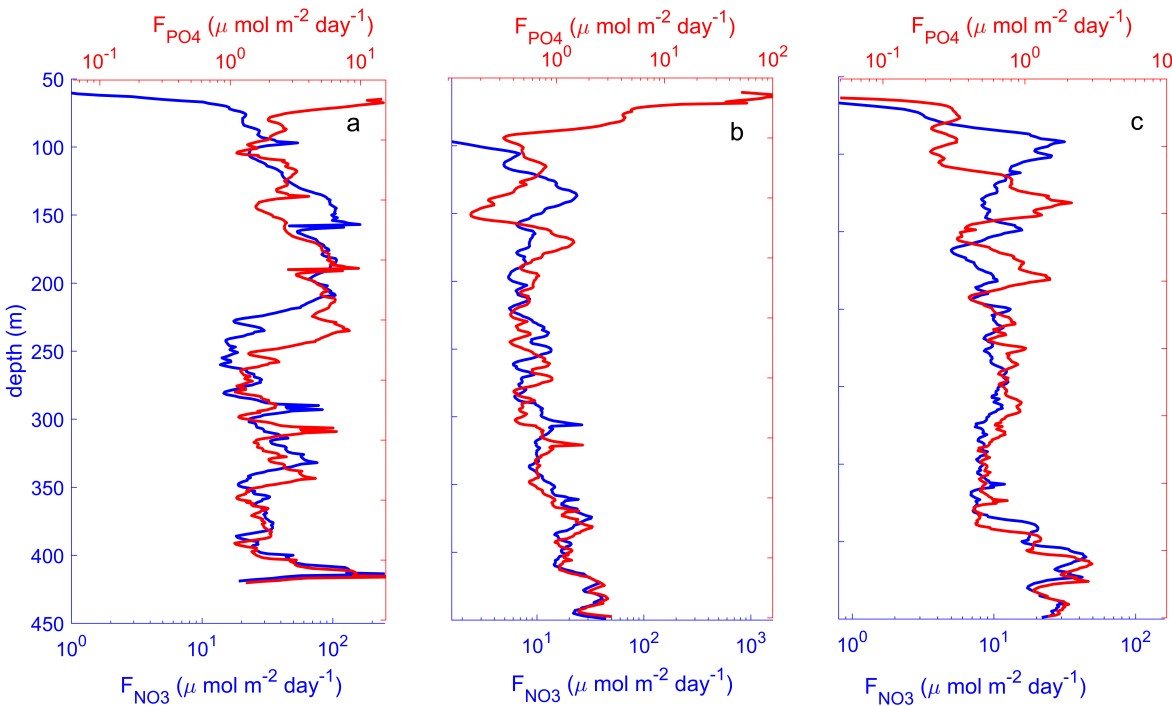

**Figure 14. Time average of vertical profiles of nitrate and phosphate turbulent diffusive fluxes at long duration stations, LD-A, (a), LD-B, (b), and LD-C, (c). The scales of the nitrate and phosphate diffusive flux are set to match with the typical Redfield ratio ($N : P = 16 : 1$).**

**Table 1.** VMP profiles.

| Station | position | depth (m) | number of VMP profiles |
|---|---|---|---|
| SD1 | $[159°54.0'E; 18°0.0'S]$ | 4068 | 2 |
| SD2 | $[162°7.5'E; 18°37.5'S]$ | 2567 | 1 |
| SD3 | $[164°54'E; 19°19.0'S]$ | 3252 | 1 |
| LD-A | $[164°41.28'E; 19°12.78'S]$ | 3491 | 30 |
| SD4 | $[168°0.0'E; 20°0.0'S]$ | 4995 | 1 |
| SD5 | $[170°0.0'E; 22°0.0'S]$ | 4405 | 1 |
| SD6 | $[172°8.0'E; 21°22'S]$ | 2509 | 3 |
| SD7 | $[174°16'E; 20°44'S]$ | 2451 | 2 |
| SD8 | $[176°24'E; 20°06'S]$ | 2028 | 3 |
| SD9 | $[178°39'E; 20°57'S]$ | 3864 | 2 |
| SD10 | $[178°31'W; 20°28'S]$ | 819 | 1 |
| SD11 | $[175°40'W; 19°59'S]$ | 2234 | 1 |
| SD12 | $[172°50'W; 19°29'S]$ | 7717 | 1 |
| LD-B | $[170°51.5'W; 18°14.4'S]$ | 4912 | 35 |
| SD13 | $[169°4.37'W; 18°12.04'S]$ | 4598 | 0 |
| LD-C | $[165°45.4'W; 18°40.8'S]$ | 5277 | 37 |
| SD14 | $[163°0.0'W; 18°25'S]$ | 3640 | 1 |
| SD15 | $[160°0.0'W; 18°16'S]$ | 3916 | 1 |

**Table 2.** Statistics in the Western and Eastern parts: percentage of $Ri < 1$ and mean values and standard deviations of $\epsilon$ and $K_z$ within a $100m - 500m$ surface layer.

| | $Ri < 1$ (%) | $< \epsilon >$ $(Wkg^{-1})$ | $\delta(\epsilon)$ $(Wkg^{-1})$ | $< K_z >$ $(m^2 s^{-1})$ | $\delta(K_z)$ $(m^2 s^{-1})$ |
|---|---|---|---|---|---|
| West of 170W | 3.0 | $2.3 \times 10^{-9}$ | $6.0 \times 10^{-9}$ | $6.0 \times 10^{-6}$ | $7.1 \times 10^{-5}$ |
| East of 170W | 0.05 | $7.0 \times 10^{-10}$ | $5.8 \times 10^{-10}$ | $2.8 \times 10^{-6}$ | $4.5 \times 10^{-6}$ |

**Table 3.** Statistics at the long duration stations: percentage of $N^2 - S^2 < 0$ (i.e. $Ri < 1$), median values of $\epsilon$, $K_z$, mean value of kinetic energy for the sub-inertial frequencies, inertial frequency and semi-diurnal M2 tidal constituent and same for shear variance. The average is performed over a within a $100m - 500m$ surface layer (first three lines) as well as over a $50m - 250m$ layer to highlight the impact of the niw at LD-A.

| | $Ri < 1$ (%) | $<\epsilon>$ $(Wkg^{-1})$ | $<K_z>$ $(m^2 s^{-1})$ | $KE_{lf}$ $(m^2 s^{-2})$ | $KE_f$ $(m^2 s^{-2})$ | $KE_{M2}$ $(m^2 s^{-2})$ | $S_{lf}^2$ $(s^{-2})$ | $S_f^2$ $(s^{-2})$ | $S_{M2}^2$ $(s^{-2})$ |
|---|---|---|---|---|---|---|---|---|---|
| LD-A | 18% | $4.4 \times 10^{-9}$ | $1.1 \times 10^{-5}$ | $2.5 \times 10^{-3}$ | $4.6 \times 10^{-3}$ | $1.4 \times 10^{-3}$ | $1.0 \times 10^{-6}$ | $8.2 \times 10^{-6}$ | $3.3 \times 10^{-7}$ |
| LD-B | 0.86% | $8.3 \times 10^{-10}$ | $3.4 \times 10^{-6}$ | $1.6 \times 10^{-3}$ | $1.2 \times 10^{-3}$ | $8.3 \times 10^{-4}$ | $1.5 \times 10^{-7}$ | $1.5 \times 10^{-6}$ | $1.0 \times 10^{-7}$ |
| LD-C | 0.25% | $6.9 \times 10^{-10}$ | $2.7 \times 10^{-6}$ | $3.4 \times 10^{-4}$ | $7.5 \times 10^{-4}$ | $7.5 \times 10^{-4}$ | $9.4 \times 10^{-8}$ | $7.8 \times 10^{-7}$ | $8.4 \times 10^{-8}$ |
| LD-A | 38.6% | $8.26 \times 10^{-9}$ | $1.0 \times 10^{-5}$ | $3.7 \times 10^{-3}$ | $1.1 \times 10^{-2}$ | $1.7 \times 10^{-3}$ | $2.1 \times 10^{-6}$ | $2.0 \times 10^{-5}$ | $7.6 \times 10^{-7}$ |
| LD-B | 0.11% | $1.7 \times 10^{-9}$ | $3.7 \times 10^{-6}$ | $1.2 \times 10^{-3}$ | $1.6 \times 10^{-3}$ | $1.1 \times 10^{-3}$ | $2.0 \times 10^{-7}$ | $1.4 \times 10^{-6}$ | $1.4 \times 10^{-7}$ |
| LD-C | 0.10% | $2.7 \times 10^{-9}$ | $3.6 \times 10^{-6}$ | $5.5 \times 10^{-4}$ | $8.8 \times 10^{-4}$ | $6.2 \times 10^{-4}$ | $2.6 \times 10^{-7}$ | $1.2 \times 10^{-6}$ | $1.1 \times 10^{-7}$ |

**Table 4.** Statistics in the Western and Eastern parts: mean values of the $NO_3$ and $PO_4$ turbulent diffusive fluxes in the photic layer. The standard deviation is given within the brackets.

| flux $(\mu mol\, m^{-2} d^{-1})$ | West of 170W | East of 170W | LD-A | LD-B | LD-C |
|---|---|---|---|---|---|
| Nitrate flux | 11.38 [19.94] | 3.15 [1.32] | 8.41 [12.40] | - | - |
| Phosphate flux | 2.66 [35.63] | 4.01 [13.70] | 2.13 [2.42] | 2.16 [5.05] | 0.31 [0.39] |

**Table 5. Relative contributions to the flux variations of $K_z$ and $c_z$ depth-averaged over a $100-m$ vertical layer defined from the top of the nitracline, West of $170W$ and East of $170W$ and at long duration stations. The mean value and standard deviation are also given.**

| | $-\Delta K_z c_z$ ($\mu mol m^{-2} d^{-1}$) | $-K_z \Delta c_z|-$ ($\mu mol m^{-2} d^{-1}$) | $<>$ ($\mu mol m^{-2} d^{-1}$) | $\delta$ ($\mu mol m^{-2} d^{-1}$) |
|---|---|---|---|---|
| West of $170W$ | +18% | -6% | 19.5 | 20.8 |
| East of $170W$ | -56% | +14% | 12.6 | 11 |
| LD-A | +105% | -16% | 44.77 | 91.77 |
| LD-B | -50% | +1% | 14.17 | 18.26 |
| LD-C | -55% | +8% | 13.1 | 15.68 |