# Peer review of "Longitudinal contrast in Turbulence along a $\sim 19^\circ S$ section in the Pacific and its consequences on biogeochemical fluxes"

_Biogeosciences, 2018_

## Referee Comment (RC1) · BFC Fernández-Castro (Referee) · 7 May 2018

REVIEW of "Longitudinal contrast in Turbulence along a âĹij19S section in the Pacific and its consequences on biogeochemical fluxes" by Bouruet-Auberot et al.

This manuscript describes a set of microstructure turbulence measurements along a longitudinal transect in the western South Pacific subtropical gyre. An interesting longitudinal gradient in both the intensity of turbulent dissipation and the mechanisms responsible for that dissipation is found. The biogeochemical implications of this gradient are also explored with the calculation of nitrate and phosphate diffusive fluxes across the base and within the photic layer. The longitudinal variability of nutrient supply was coherent with an increasing degree of oligotrophy to the east and the different degree of penetration of the phosphate and nitrate fluxes into the surface ocean was related to the activity of nitrogen fixers. The dataset presented in the manuscript fills a substantial gap of knowledge, providing microstrucutre measurements in a largely undersampled area. The analysis of the turbulence generation mechanisms is promising and the implications for the biogeochemistry are very interesting. However, I have found a number of problems in the manuscript, mainly related to the presentation and discussion of the results and to the lack of some important information in the methods section. In my opinion the manuscript is not suitable for publication in the present form and a major revision would be required.

GENERAL COMMENTS:

- The text needs to be revised in order to improve the communication of the results. Many parts of the manuscript are difficult to follow or contain grammatical and typographic errors. Sections 3 and 5 are structured in single extremely long paragraphs and are very difficult read. These sections need to be restructured and split into paragraphs. Some of the figures contain errors, for example wrong axis labels (Figure 11)

-There is general lack of methods information. In particular, I was not able to find a description of the method used to obtain the tidal forcing (Figure 5) and, more strikingly, there is no explicit mention to how the nutrient fluxes were computed. In this sense, one of the most outstanding results from a biogeochemical perspective is that, contrary to nitrate, the phosphate upward diffusive flux does not drop to zero above the DCM. This is probably because the nitrate gradient above this depth is virtually "zero" but the nutrient distributions are not shown and the calculation method is not reported. I guess that there is some "noise" in both the nitrate gradients and the Kz and that the fluxes are not actually "zero" but below some "noise" level. How was "zero" defined? Another important point is that, in my opinion, the use of a fixed averaging interval (20-80 m) for the calculation of the nutrient fluxes within the photic layer (Figure 14) is not the best choice because the photic layer dimensions change with longitude. From my point of

view, the photic layer fluxes should be calculated in a depth range consistent with this variability, as done for the fluxes across the nitracline.

-In general I miss more quantitative information in the text. The description of the results is mostly based on the qualitative description of the figures. I believe that reporting some quantification of the average TKE dissipation rates and nutrient fluxes in the different situations/regions (namely in sections 4 and 5) would help to structure the text and communicate the results more effectively. In the particular case of section 3, I would suggest to define a separation between the eastern and western parts of the section based on longitude (e.g. at 190°E) or bathymetry and obtain some statistics for the different parameters in both parts (eg. mean or median values of epsilon, K, percentage of subcritical Ri bins, etc.). Also, the results from LD stations highlighted the impact mixing intermittency, with implications for biogeochemical fluxes. I find this information novel and very valuable, and it could be better illustrated with some numbers/statistics. The quantification of the N:P ratios of the diffusive nutrient fluxes across the nitracline and within the photic layer could also be helpful for the discussion the biogeochemical implications.

-One of the main focus of the manuscript is to demonstrate that the spatial patterns of dissipation rates are related to the west-east gradient in the intensity of internal wave generation (and dissipation). However, in the first part of section 3 (lines 4-17 of page 5) and in the conclusions, the authors mention shear instability as a possible driver of the longitudinal asymmetry, based on the distribution of the Ri numbers along the section. It is not entirely clear to me whether the authors want to suggest that the presence of subcritical (and patchy) Ri derives from a mechanism other than internal waves (i.e. low frequency flow), as it seems to be pointed out in the conclusions. I think this point needs some clarification and better justification. The authors could add some more insights to the discussion of Figure 10 or include a similar decomposition for shear variance.

SPECIFIC COMMENTS:

Page 2, lines 24-27: "Global maps of the energy flux into inertial motions is enhanced at mid-latitudes as well as around a SE oriented track from the Equator to 40S and within âĹij $180 - 200E$ longitude (Alford and Zhao, 2007, Fig.9)". The authors may specify whether "mid latitudes" refers to the south Pacific basin in particular or to the global ocean.

Sometimes the authors refer to inertial waves and others to near-inertial waves. This is confusing to me. For example, in this sentence (Page 1, line 25)" Global maps of the energy flux into near-inertial motions show enhanced semi-diurnal tide energy conversion in the western part of the subtropical South Pacific [...]" it seems that you are referring to the M2 internal tide as near inertial. At a latitude of $\sim20°S$, the inertial period if about 35 hours (if I am no wrong). Is it correct to say that semi-diurnal internal tides are near-inertial?

Page 3, lines 3-5: "The purpose of this paper is to characterize three-dimensional turbulence along the OUTPACE transect with microstructure measurements performed at both one-day short duration stations and at long duration stations lasting three inertial periods." I would say that you characterize the "spatial variability" of microstructure turbulence rather than three-dimensional turbulence. (It also applies to the abstract)

Page 3, Section 2: were all the instruments (CTD, VMP) deployed in all stations? Is LADCP data used in this manuscript? I could not find it in the figures or text.

Page 3, line 17: add reference for the Visbeck inversion method

Page 3, section 2.2. Indicate the approximate maximum depth of the microstructure sampling. Indicate the approximate number of profiles in long and short duration stations, eg. $\sim30$ and 1-3 profiles.

Page 3, line 29. More detailed information about the microstructure data processing would be desirable. For example, how was the noise level estimated? From which depth were the epsilon data considered reliable? Was there any noise removal proce-

dure applied? How was the information from the two shear sensors merged?

Page 4 line 24-25, indicate the approximate depth range for epsilon and Kz averaging as in Fig. 1 caption (∼100 to 800 m). What was the mixed layer depth? You could add the distribution of MLD to Figure 3.

Pages 4-5, section 3: This section is described in a extremely long single paragraph. I would suggest to split the section into 3-4 paragraphs to facilitate the reading

Page 5, line 5-6: " More insights on turbulence are given with vertical sections of epsilon and Kz in Figure 3a and b." What is represented in this figure, station-averaged profiles?

Page 5, line 6: This sentence: "The range of epsilon values covers 3 orders of magnitude, typically below the mixed layer down to 300m depth, and presents a typical patchy pattern with spots of intense turbulence with values up to âLij $10-8$ W kg$-1$ down to 500m. Most of these events are observed in the West and their occurrence decreases eastward and downward (Fig.3a)." is not clear to me. Please be more precise.

Page 5, lines 12-13. Add °E. I don't appreciate the absence of a marked pycnocline in 180 and 180°E. Could you explain better? In general, the description of the vertical and horizontal distributions of N, S and Ri could be improved. Sometimes it is difficult to know for sure to which depth intervals the authors are referring.

Page 5, lines 18-19: How was this information obtained? From which source? A more thorough explanation of this analysis is definitely required, similar to that given for the input of near-inertial energy (lines 25 to 29).

Page 5, line 25. What is the inertial period in the study area?

Page 6, line 21. Figure 9 is abruptly introduced here without any specific explanation. Panels (a) -(c) are not mentioned at all. On the other hand, panels (d)-(i) introduce redundant information already present in Figure 6. In my opinion the authors could just drop this figure and sustain their argumentation with Figure 6, which is already familiar

for the reader.

Page 6, line 28 onwards and Figure 10: How were the energies associated with the different frequencies calculated? A similar frequency-decomposition of the shear variance could be useful to better separate the processes contributing to shear instability (internal waves vs. low frequency) [*]

Page 6, line 31: "[...] a wave mean flow interaction (i.e. critial level)." Perhaps a reference is needed here

Page 7, line 4-5. "The contrast in turbulence between the three stations is mostly confined in the upper few hundred meters as a result of an energetic niw and its interaction with the strongly sheared subinertial flow." Are you refering to Figure 10a where you can see a decrease of low frequency energy with depth? A direct quantification of shear variance in the different frequencies could help to visualize this. See previous comment [*]

Page 7, section 5. The nutrient distributions are not shown and the sampling and methodological details are not reported in the manuscript. You must at least provide a reference where this information can be found. The methodology used to calculate the diffusive fluxes is not reported either. Were the VMP and nutrient sampling vertical grid coincident? Was some interpolation required to match the vertical resolution of both variables? Again this section is too long to be written in a single paragraph.

Page 7, line 14: "Large variations are noted, that result from the strong variability of Kz (Fig.11b).". Specify that these variations are in the "short-scale" in contrast with the large-scale longitudinal gradient.

Page 7, line 17. The authors may state that the nitrate flux is zero above the DCM because the gradient/concentration is zero.

Page 7, lines 17-20. change "[...] of the nitrate diffusive flux within the Redfield ratio [...]" to "[...] of the nitrate diffusive flux by a factor of 1/6 corresponding to the Redfield

ratio [...]". I think "followings" is not correct in English.

Page 7, line 30-33. What is the euphotic layer depth and how does it relate to the nitracline? You could show the calculation interval in Figure 13

Page 8, line 1: "The mean nitrate turbulent diffusive flux is far larger [...]" What is the mean flux? "Far" is not quantitative. Give some numbers

Page 8, lines 5-6: give numbers

Page 8, lines 7-10. From your data I would not say that the nitrate flux into the photic layer is negligible in the Malasian Archipelago (LD-A). The depth of the photic layer is usually some meters below the DCM which is located at ~80-100 m in LD-A. At this depth the nitrate fluxes are not zero (Figure 12c). If it is negliglible in comparison with N2 fixation, could you give some typical value of N2 fixation rate to compare.

Page 8, lines 10 – 15: According to your data, the nitrate flux vanishes above the base of the euphotic zone and the phosphate flux reaches shallower depths, potentially fueling nitrogen fixation. I find this result very interesting. Now this question raises to me: is the supply at the base of the DCM Redfieldian (N:P ~ 16), and, thus, net production at the DCM results in a preferential uptake of nitrate (N:P>16), such that the nitrate flux gets exhausted first, or, on the contrary, the nutrient supply is already nitrogen-depleted at the base of the DCM, i.e. the N:P ratio of the diffusive flux at the nitracline is <16? It is just for my personal curiosity, but it might also be interesting to discuss that in the manuscript. You could show the phosphate fluxes as well in Figure 13 and report the mean N:P values in the text. You could compare these N:P ratios with those at shallower depths (Figure 14).

Page 8, lines 16-28: In my opinion the a choice of a constant interval for the flux integration within the photic layer is not the best choice here because the different stations exhibit different photic layer depths, with an eastward deepening of the DCM. The use of a fixed interval results in zero nitrate fluxes in LD-C, but not in the others.

This might be reflecting only the different dimensions of the system but not substantial differences in nutrient cycling dynamics. The authors might refer the lower limit of the interval to the depth of the top of the nitracline or the (upper) DCM, as in Figure 13.

Page 8, lines 22-24."While at LD-A the phosphate turbulent diffusive flux is of the same order of magnitude as that of the nitrate turbulent diffusive flux at LD-A (Fig.14b and c) there is at least an order of magnitude difference between phosphate and nitrate turbulent diffusive fluxes at LD-B (Fig.14e and f).". The comparison between the nitrate and phosphate fluxes would be better done in terms of the Redfield ratio, otherwise it is confusing.

Page 9, lines 1-7. It is not entirely clear to me if you suggest that the shear instability mixing, based on the distribution of the Ri number along the transect, derives from a mechanism other than internal waves, i.e., strongly sheared mean sheared currents as you seem to point out here. Your Figure 10 indicates that the most energetic currents correspond to the semidiurnal and inertial periods, with a generally minor contribution of the low frequencies, at least in the upper 400-500 m. Is it possible that the patchy Ri patterns derive from internal waves becoming shear-unstable and not due to shear in the mean currents? The separation between the two processes is not sufficiently argued, from my point of view. See a previous comment [*]

Same lines: There is an extensive work on shear-driven equatorial turbulence by W. D. Smyth, J.N. Moum and colaborators. The authors could possibly include some reference to their work. Eg: "Smyth, W. D., Moum, J. N., Li, L., & Thorpe, S. a. (2013). Diurnal Shear Instability, the Descent of the Surface Shear Layer, and the Deep Cycle of Equatorial Turbulence. Journal of Physical Oceanography, 43(11), 2432–2455. https://doi.org/10.1175/JPO-D-13-089.1" or "Smyth, W. D., & Moum, J. N. (2013). Marginal instability and deep cycle turbulence in the eastern equatorial Pacific Ocean. Geophysical Research Letters, 40(23), 6181–6185. https://doi.org/10.1002/2013GL058403"

[Figure]

Page 9, lines 22-23: "Phosphate turbulent diffusive fluxes mean values were significant in the euphotic layer with the exception of the most eastern station." What does "not significant in the eastern most station" mean? What are the confidence intervals?

TECHNICAL COMMENTS

Page 1 Title and throughout the manuscript: add degree symbol to 19S. Sometimes "S" and "E" are shown in italics, which I believe is not correct.

Page 1, line 7: What does "surface layer" mean here. The longitudinal differences in turbulent dissipation reach ∼400 m. I would not call this a "surface layer"

Page 1 Line 14: Averaged nitrate turbulent diffusive fluxes *ACROSS THE BASE OF THE PHOTIC ZONE* were at least twice as large at the western station than at the two eastern stations due to the *LARGER* vertical diffusion coefficient.

Page 2, Line 14: I would rather start a new paragraph after "Ledwell et al., 2008"

Page 2 Line 27: There is no Figure 9 in "Alford, M. H. and Z. Zhao, 2007: Global patterns of low-mode internal-wave propagation. part ii: Group velocity. Journal of physical oceanography, 37 (7), 1849–1858." Is this the correct reference? I believe the authors intended to refer to "Alford, M.H. and Z. Zhao, 2007: Global Patterns of Low-Mode Internal-Wave Propagation. Part I: Energy and Energy Flux. J. Phys. Oceanogr., 37, 1829–1848, https://doi.org/10.1175/JPO3085.1"

Page 2, Line 27 and throughout the manuscript: the format of the references to the figures is incoherent. Many different formats are used, eg. Fig.9 (Line 27), Fig. 1 (Line 12), Fig6d (Line 33). Please uniformise.

Page 2, Line 31: purposeS

Page 2, Line 32 and throughout the manuscript: in "N2 fixation", "2" should be subscript as in Page 1, Line 16

Page 2, Line 33. Italics: Trichodesmium

Page 3, line 19 and throughout the manuscript: there is no space between units and the corresponding figures (eg. 2min). I would suggest to add a space here

Page 3, line 29 and throughout the manuscript: Units should not be in italics

Page 4 line 20: the molecular viscosity was already defined in line 11, move "$\nu = 1.2 \times 10-6$ m2 s$-1$." to line 11.

Page 4, Line 26: "...West of 185°E"

Page 5, line 10: end paragraph here?

Page 5, line 17: end paragraph here?

Page 5, line 21: remove "...of our study area"

Page 5, lines 29-30: In this sentence "The maps reveal a striking longitudinal contrast in inertial flux until mid March (Fig.6a-e)", striking might be too strong.

Page 6, line 9. In "... the shear is far larger" remove far*

Page 6, lines 6-7: In the sentence "Turbulence at LD-A is by far the largest down to 400m depth with contrasted mean epsilon and Kz between LD-A on one hand and LD-B and LD-C on the other hand (Fig7a and b), within a factor of 5−10 for epsilon and Kz." the authors intended to describe both the vertical distribution and the variability between stations, which makes the reading and interpretation very difficult. Also, "by far" is imprecise here. I suggest to split the sentence into sentences and report some mean epsilon or Kz values to better quantify the differences. I am not a native English speaker but I have the impression that it would be better to use "by a factor of 5-10" instead of "within a factor of 5-10". At least it is easier to understand, from my point of view.

Page 6, line 28. Maybe change to "The enhanced epsilon at LD-A is *coincident with an energetic niw *at 50-200 m (Fig.10a)."

Page 8, lines 2-4. This sentence is too long. Consider splitting.

Page 8, line 14: I can't figure out the meaning of "locally" in this sentence

Page 8, line 16. Consider to introduce a new paragraph here.

Figure 1: -Add epsilon and K_z symbols in the caption. Eg. "Log values of dissipation rate of turbulent kinetic energy (epsilon, W kg $-1$ )". -Remove "(log scale)", this information is repeated. -Add "longitude ($^{\circ}$E)" in the xlabel. The same in the following figures -The caption states "Time-averaged values at long duration stations, LD-A, LD-B and LD-C are displayed with diamonds while values at short duration stations are displayed with circles.", however, I could not see any diamond in this figure

Figure 2: -Magenta symbols and lines are not easily visible for me in this figure (and others). I would suggest to use a different color - In the Methods sections the reported SADCP frequencies are 150 and 75kHz. According to the Figure caption velocity data were obtained with a 38kHz SADCP. Is it a different instrument?

Figure 3: -I would suggest to represent the mixed layer depth -Indicate whether the represented profiles are station-averages or individual profiles -Circles overlap with each other more than I would like to. In this way it is difficult to interpret the vertical patterns. I would suggest to make the figure larger in the vertical dimension in order to reduce the overlap.

Figure 4: -Panel c: the authors could highlight somehow the Ri values <1 or <0.25, to stress the areas of instability. If the information is the same as represented in Figure 3, you could also use the same color scale to avoid confusion.

Figure 7: - Caption: specify with which instruments N2 and S2 were obtained. add something like that:"[...] were inferred from the rosette-mounted CTD and LADCP instruments/SADCP(?)"

Figure 10: -Could you specify to which SADCP each line corresponds in the legend as well?

Figure 11: -The x-scale of the subplots is different. I am also confused by the number of profiles shown in panels (b-c). There are more profiles shown here than stations in the cruise (18) but less than the total number of profiles (>100). How is that possible? Are they station-averaged profiles? If not, what does the x-axis represent? -What do the shaded areas represent? Zero vertical gradient (= zero flux)?Indicate

Figure 12: -Caption: "Longitude depth sections of ..." Longitude-depth is not correct. Does not the x-axis represent time in days as in Figure 8?

Figure 13: -Add phosphate fluxes. You could also add mean (or median) values and confidence intervals

Figure 14: -You could add mean (or median) values and confidence intervals

---

## Referee Comment (RC2) · Anonymous Referee #2 · 22 May 2018

Overview: The manuscript describes an impressive set of turbulent microstructure data in an attempt to assess near-surface turbulent mixing in a generally oligotrophic region and the main driving mechanisms for the turbulence (inertial or internal tide shear). The measurments are used to estimate the supply of nitrate and phosphate to the euphotic zone, with some interesting consequences suggested by non-Redfield fluxes in the surface layer. The data is very strong, and the aims are novel and important. My main suggestions focus on some more quantitative analyses to support the claims made.

General points: A key aspect of the quantitative analyses of the turbulence data is the

demonstration of turbulent dissipation alongside regions of close-to-critical Richardson number, supporting the suggestion that turbulence was generated by shear instability. (e.g. Page 5 line 16: reference is made to subcritical Ri). This is really difficult to see in Fig. 3. The coloured dots overlap considerably, which makes the profiles of dissipation and diffusion difficult to see. The shading of Ri does not really convey useful information. Assuming the critical Ri is being taken to be about 1, then it needs to be clear where that is. The log scale makes Ri=1 roughly white I think, but I cannot see what the text on page 5 discusses. Why does Ri use a log scale? You are really only interested in changes in Ri about the value of 1. Why not do a more quantitative analysis – for instance, what does a scatter plot of turbulent dissipation versus Ri look like? The evidence as presented is not a convincing case for shear instability. Perhaps one of the most interesting an important analyses is that relating turbulent dissipation to the energy in the subinertial flow at inertial and semi-diurnal frequencies (pages 6 and 7). However, this analysis lacks any real quantitative evidence. It is based largely on a qualitative comparison of vertical profiles in Figure 10, which is not adequate in supporting the assertions made on the drivers for turbulence (particularly as an important suggestion is that the higher dissipation in the west is not driven by the most obvious candidate of rougher seabed topography and more internal tidal activity). This analysis needs to be strengthened. Also, the dissipation profiles in Figure 10 (and Fig. 7) would benefit greatly from having the 95% uncertainties added alongside the mean profiles (e.g. bootstrapping the profiles at each station – there is plenty of them), which would better highlight just how strong the contrasts are between the stations.

Specific edits and smaller suggestions/queries: 1. The title should really be "…along a 19ïĆřS section….." 2. Line 2 in the Abstract, if it is necessary to have the Moutin & Bonnet reference here, then include the complete reference. 3. Line 3 Abstract: "…hydrographic and current measurements at fine scale…". What is meant by fine scale? The horizontal spacing of the CTD profiles could not really be described as "fine", and while the vessel ADCP data could be at fine scale, it is not used as such. 4. Line 6 abstract: "….with stronger turbulence in the west, i.e….." 5. Line 8 abstract:

"….pattern was correlated with the energy…." 6. Line 13/14 abstract: Turbulent nitrate fluxes are described asgreater in the west because of the increase in eddy diffusivity. What proportion of the increase was because of Kz, and what caused by changes in the nitrate gradient? 7. Line 16 abstract: "….organisms that were seen to be the main contributors…." 8. Page 2 line 1: "…increasing oligotrophy to the East." Presumably "…increasing oligotrophy towards the east and the centre of the gyre"? Oligotrophy would lessen if you kept going east…. 9. Page 2 line 18: The dissipation is reported for the "stratified ïА¿300m". I am not sure what this means. Is it an average dissipation over the upper 300 metres? 10. Page 2 line 31: "the main purposes of…" 11. Page 2 line 33: trichodesmium should be Trichodesmium (capital letter and italics). 12. Page 2 line 34: "…turbulent diffusion was found to make a negligible…." 13. Page 3 line 1: "leads to the question of the sources of other nutrients to the euphotic layer that could sustain…." 14. Page 3 lines 4/5: "The aim is also to provide insight into the main mechanisms…." 15. Page 3 line 6: "…dynamics influence biogeochemical…" 16. Page 3 line 9: French (capitalised). 17. Page 3 line 10: the short duration stations are described as "24 hour", and are later described as having "a few profiles" (line 24) of microstructure. Most of these short stations only had 1 profile, which I assume took a lot less than 24 hours and does not count as a "few". This should be clarified. 18. Page 3 line 18/19: "…yielding processed currents…" 19. Page 3 line 27: "…which allowed validation of the estimate…" 20. Page 3 line 28: dissipation is described as being calculated in 1 metre bins and then averaged over 8 metres. Is this a standard analytical procedure? 21. Page 3 line 29: "level is 5…" 22. Page 4 line 7: was N also calculated on 1 metre bins before the 8 metre averaging? 23. Page 4 line 8: "ïАĞ has generally been set to…recent findings of Shih…." 24. Sections 3, 4 and 5 each constitute Results and Discussion on 3 different topics. I suggest use a general section 3 Results and Discussion, and then subsections 3.1 Spatial pattern of Turbulence, 3.2 Possible impact of internal waves, etc. The section on Spatial pattern of turbulence is in need of splitting into coherent paragraphs – at the moment it is a fairly dense section of text that makes it hard work for the reader (well, at least this reader). 25. Page

4 line 24: by "depth averaged" below the mixed layer, I assume you mean average between the base of the mixed layer (how defined?) and the deepest reached by the profiler? 26. The longitude axes of the data continues to increase in value past 180ïĆřE. I know this makes life much easier for plotting the data, but fundamentally it is not the correct horizontal coordinate system. Specify in terms of correct longitude, and include "degrees" or "ïĆř". 27. Page 5 line 11: how was shear, S, calculated? From the LADCP or SADCP? What depth bins? What time averaging (i.e. how many raw profiles)? 28. Page 7, lines 13,14. The changes in vertical nitrate flux are attributed entirely to changes in Kz. This looks reasonable, based on Fig. 11, but are there any changes in the strength of the vertical nitrate gradient that might also contribute? 29. The caption to Fig. 12 is incorrect. 30. Fig. 13. Is the dashed line in each panel the mean value? The top of the nitracline has been defined I assume on the basis of an interval with one end pinned by nitrate reaching undetectable concentrations. If the euphotic zone were defined in terms of the 1% irradiance, would that change the results. 31. It would be useful to provide some context for the values of the nitrate flux measured – how do they compare with other published values (e.g. Planas et al., Limnol. Oceanogr. 1999, 44, 116-126; Lewis et al., Science, 1986, 234, 870-873; Stevens et al., Limnol. Oceanogr. 2012, 57, 897-911).

---

## Author Comment (AC1) · 24 Oct 2018

see all documents in the zip file below

Please also note the supplement to this comment:
https://www.biogeosciences-discuss.net/bg-2018-170/bg-2018-170-AC1-supplement.zip

---

## Author Comment (AC2) · 24 Oct 2018

see all documents in the zip file below

Please also note the supplement to this comment:
https://www.biogeosciences-discuss.net/bg-2018-170/bg-2018-170-AC2-supplement.zip

---

## Author Response (AR1)

**LABORATOIRE D'OCEANOGRAPHIE ET DU CLIMAT :**
**EXPERIMENTATION ET APPROCHES NUMERIQUES**

*UNITE MIXTE DE RECHERCHE 7159*
*CNRS / IRD / UNIVERSITE PIERRE & MARIE CURIE / MNHN*

*INSTITUT PIERRE-SIMON LAPLACE*

[Figure]

Dear Editor and Reviewers,

Thank you for your feedback on the revised version of our manuscript. The minor revisions have been performed and the manuscript download as well as the tex file and figures.

Best regards

Pascale Bouruet-Aubertot

*Université Pierre & Marie Curie - boîte 100 - 4, Place Jussieu - 75252 PARIS CEDEX 05Tél. : +33 1 44 27 32 48 - Télex : 2063 17 F -*
*e-mail : pascale.bouruet-aubertot@locean-ipsl.upmc.fr - web : http://www.locean-ipsl.upmc.fr*
*Direction et administration : Tour 45-55, 4eme étage – fax +33 1 44 27 38 05*